# Towards Evaluating Generalist Agents: An Automated Benchmark in Open World

## Abstract

Evaluating generalist agents presents significant challenges due to their wide-ranging abilities and the limitations of current benchmarks in assessing true generalization. We introduce the **M**ine**C**raft **U**niverse (**MCU**), a fully automated benchmarking framework set within the open-world game *Minecraft*. MCU dynamically generates and evaluates a broad spectrum of tasks, offering three core components: 1) a task generation mechanism that provides high degrees of freedom and variability, 2) an ever-expanding set of over **3K** composable atomic tasks, and 3) a general evaluation framework that supports open-ended task assessment. By integrating large language models (LLMs), MCU dynamically creates diverse environments for each evaluation, fostering agent generalization. The framework uses a vision-language model (VLM) to automatically generate evaluation criteria, achieving over 90% agreement with human ratings across multi-dimensional assessments, which demonstrates that MCU is a scalable and explainable solution for evaluating generalist agents. Additionally, we show that while state-of-the-art foundational models perform well on specific tasks, they often struggle with increased task diversity and difficulty.

## 1 Introduction

In recent years, large language models (LLMs) have demonstrated remarkable progress in the field of AI (Touvron et al., 2023; Achiam et al., 2023). The release of the GPT series (Brown et al., 2020) has significantly reshaped AI research, moving the focus away from task-specific models toward the development of foundation models. (Bubeck et al., 2023). These models excel across a diverse set of tasks and are highly instructable, marking a substantial leap forward in versatility and adaptability. The next step in this evolution is the development of *Generalist Agents* (Bubeck et al., 2023). So, what is a Generalist Agent? From the perspective of users, the ideal generalist agent should embody a multifaceted utility, seamlessly integrating a spectrum of complex services. For instance, users typically prefer asking ChatGPT for a range of services like searching, translation, writing, coding, etc., rather than relying on numerous specialized apps. This preference underscores the potential for a "single-brain" style generalist agent, which intriguingly aligns with neuroscience insights (Mountcastle, 1978; Zhu et al., 2020; Taylor, 2005), offering a two-way benefit. Beyond that, generalist agent extends its capabilities by being able to interact with its environment, directly influencing and adapting to the real world. This interaction capability bridges the gap between passive task execution and active decision-making in complex, dynamic settings (Reed et al., 2022; Durante et al., 2024; Oertel et al., 2020). Therefore, we think that generalists should have following two characteristics: 1) possess the generalization capability to manage diverse tasks; and 2) exhibit robust interactivity and adaptability in the real-world challenges.

Creating a generalist agent presents significant challenges. Early efforts attempted to create a "one-fits-all" network (Schmidhuber, 2018) with life-long learning strategies but struggled with basic tasks due to catastrophic forgetting (McCloskey & Cohen, 1989). Recent meta-reinforcement learning (meta-RL) studies (Finn et al., 2017; Hospedales et al., 2021; Lake & Baroni, 2023) has shown potential in endowing models with human-like abilities for systematic generalization, but challenges such as scalability, sample inefficiency, and limited performance in complex environments persist (Parmar et al., 2023; Hospedales et al., 2021). Recent efforts have shifted towards pretraining large foundation models on extensive internet-scale datasets (Cai et al., 2023b; Baker et al., 2022), achieving significant advances in tackling more complex and diverse tasks in open-world environments. However, these

Table 1: Comparison between MCU and related benchmarks for testing generalization

| Benchmark | Environmental-level | | | Task-level | | Evaluation-level | |
|---|---|---|---|---|---|---|---|
| | Open-world | Procedure generation | Dynamic task generation | Task Verification | Task composability | Tunable difficulty | Auto eval open-ended task |
| DmLab (Beattie et al., 2016) | × | × | × | ✓ | × | ✓ | × |
| Procgen (Cobbe et al., 2020) | × | ✓ | ✓ | × | × | ✓ | × |
| Crafter (Hafner, 2021) | ✓ | ✓ | × | × | × | × | × |
| Xland (Team et al., 2021) | ✓ | ✓ | × | × | × | × | × |
| DYVAL Zhu et al. (2023a) | × | ✓ | ✓ | ✓ | ✓ | × | × |
| Minedojo Fan et al. (2022) | ✓ | ✓ | × | × | × | × | ✓ |
| MCU (ours) | ✓ | ✓ | ✓ | ✓ | ✓ | ✓ | ✓ |

models exhibit strong performance only on a constrained set of tasks, leaving their true generalization capabilities unproven.

In light of these challenges, the need for rigorous evaluation methods becomes apparent. While benchmarks like DmLab-30(Beattie et al., 2016) and Procgen(Cobbe et al., 2020) have made strides with multi-tasks learning and procedural generation, they fall short in assessing agent within competitive environments (Stanley et al., 2017; Parmar et al., 2023). Minedojo(Fan et al., 2022) and Crafter(Hafner, 2021), have pushed forward in open-world contexts, they lack sufficient task dynamism and verification mechanisms. Other works(Zhu et al., 2023a; Zhou et al., 2020) push boundaries with dynamic task generation and composition, yet constrained by text-only modality of the tasks. The CRAB framework Xu et al. (2024) introduces a cross-environment benchmark that leverages multimodal language models to perform tasks across various GUI environments. However, the above benchmarks often face limitations in evaluating open-ended tasks due to the absence of clear completion signals, making it difficult to test agents on more creative and adaptive challenges. A comparison of these benchmarks is provided in Table 1.

To address these limitations, we introduce our benchmark, **M**ine**C**raft **U**niverse (**MCU**), which offers high degrees of freedom in task design and evaluation. Minecraft, as an open-world platform, provides a rich and diverse set of challenges, including tasks such as Trade (logical reasoning), Mining (physical interaction), Combat (strategic planning), Building (artistic creation), Trapping (precision control), and Redstone (complex-knowledge application). This variety provides agents with ample opportunities to explore and learn across diverse scenarios. At the task level, we collect over 3000 atomic, composable tasks, with the potential to infinite expansion. By leveraging large language models (LLMs), each task is dynamically generated and uniquely instantiated during each evaluation, promoting essential generalization skills in agents. Tunable difficulty is also involved to ensure more flexible testing. Furthermore, we propose a domain-general, vision-language model (VLM)-based evaluation method capable of assessing open-ended tasks, even those without explicit end signals. Crucially, our method automates the whole pipeline of task generation, verification, and evaluation, enabling scalable benchmarking (Figure 1), which paves the way for comprehensive evaluation of generalist agents. We adhere to the criteria outlined in Section2 to develop our benchmark.

## 2 BENCHMARK DESIDERATA

Based on the aforementioned challenges, we argue that three keystones should be introduced to benchmarking generalist agents.

**First, diversity is the key**. The emergence of human-like general intelligence is inextricably tied to diverse environments (Taylor, 2005). Environmental diversity drives evolutionary pressures, fostering the development of complex cognition, technological innovation, and adaptability (Elmqvist et al., 2012; Zhu et al., 2020). Similarly, diverse challenges stimulate the capacities of agents, pushing them to generalize and perform across a wide array of tasks and conditions. However, in reality, their capabilities are vastly different. In our MCU benchmark, we incorporate two types of diversity: 1) **intra-task diversity**: Each task should embody a high degree of variability and randomness, providing freedom to truly test the agent's adaptive skills. 2) **inter-task diversity**: The benchmark should encompass a broad spectrum of task categories, representing the diverse challenges agents are likely to encounter in real-world environments.

**Second, task quality deserves attention**. As the demand for automatic generation grows, some approaches (Cheng et al., 2024; Fan et al., 2022), rely heavily on large language models (LLMs)

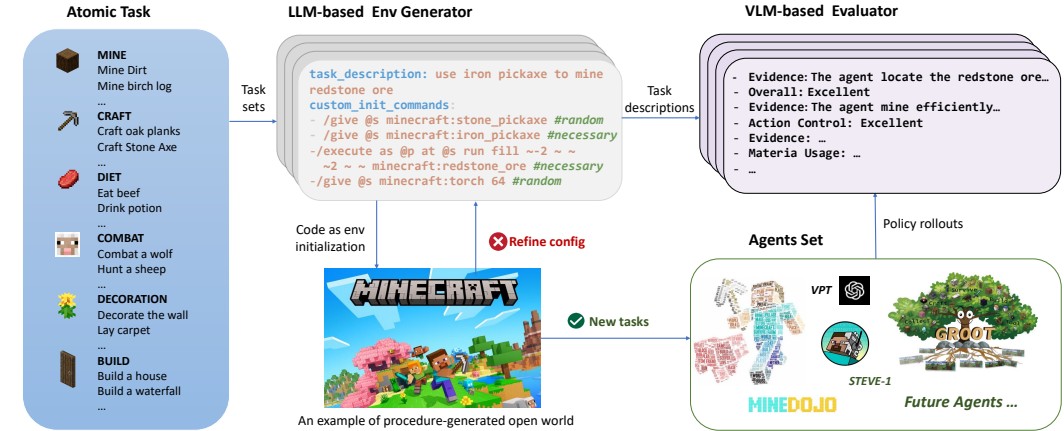

Figure 1: Overview of MCU automated benchmarking pipeline.

or procedural methods to generate numerous tasks and their corresponding initial conditions, yet it remains questionable whether these initial conditions can actually lead to the task's solution Yang et al. (2024). For instance, a task such as "mine diamond" cannot be completed with wooden pickaxe. Hence, we introduce a task generation approach based on soft constraints and a verification pipeline. Although we cannot guarantee that every task can be solved, we can ensure that more than 95% of the tasks are solvable.

**Third, an automatic evaluation system is indispensable** for fostering the development of generalist agents. Open-ended tasks (Stanley et al., 2017; Standish, 2003), by their very nature, lack well-defined end states or straightforward success signals, necessitating reliance on **human evaluation or handcrafted metrics**, which are labor-intensive and time-consuming(Dubois et al., 2024). Therefore, automatic evaluation systems that enable the large-scale evaluation of generalist agents across complex, open-ended tasks is required.

To make our automatic evaluation effective, we meet the following two criteria: 1) evaluations must be **reliable**, providing accurate assessments that align closely with human judgments. This requires the system to identify the key points of task completion, ensuring that the results are both consistent and interpretable; 2) evaluations are **multi-dimensional**. Beyond success rates, which only capture a binary measure of task completion, we need more granular such as overall skills, task efficiency, error correction, and fine-grained control of actions.

## 3 THE AUTOMATED BENCHMARKING PIPELINE

In this section, we will introduce our benchmarking pipeline. To achieve *diversity* in section2, we adopt Minecraft, an **open-world environment**, as our platform and propose an **automatic task generation** method to maximize task randomness. To ensure task quality, we define **atomic tasks** and introduce an **automatic verification** method to guarantee the solvability of the tasks. In order to conduct large-scale task evaluations, we propose an **automatic evaluation** method to alleviate the burden on humans and provide multi-dimensional assessment metrics.

### 3.1 MINECRAFT AS AN OPEN-WORLD ENVIRONMENT

For human player, there is not a pre-defined goal in Minecraft. For example, players are allowed to mine ores, craft items, build architectures, combat enemies, explore freely in the varied world with diverse biomes. Previous researches proposed classical tasks such as *Obtain Diamond* (Guss et al., 2019a) and *Find Cave* (Milani et al., 2023), but the possible tasks are endless which makes the multi-task evaluation insufficient. Furthermore, the broad open-ended tasks cover a wide spectrum of challenges in AI research, such as long-horizon decision making (Jin et al., 2023), precise control (Zhang et al., 2020), OOD generalization (Yang et al., 2023).

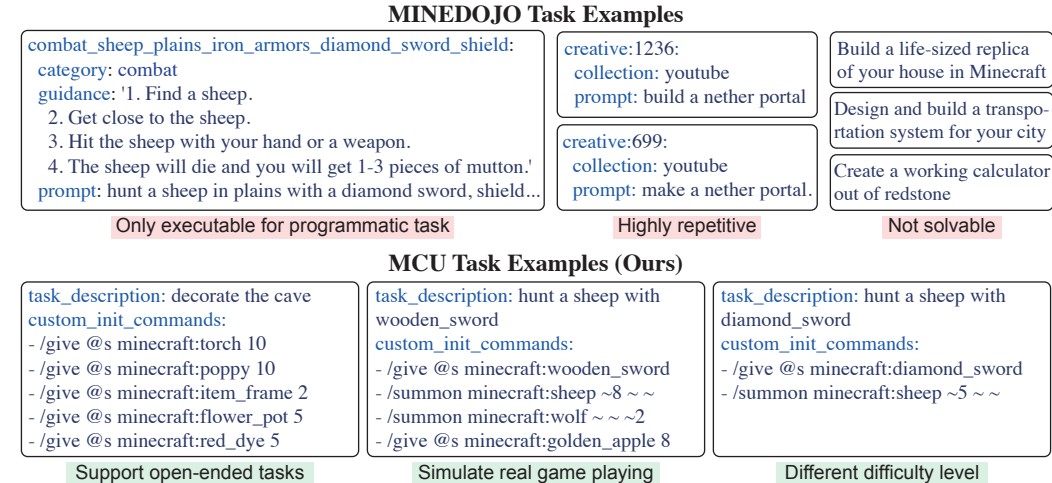

Figure 2: A comparison between the "tasks" in our MCU and Minedojo (Fan et al., 2022). We investigate the task list provided by Minedojo[2] and identify several issues. For example, only programmatic tasks that have clear reward signal can be executable in the benchmark; many tasks in their list are repetitive (both No.1236 and No.699 are "build nether portal"); and a large amount of tasks in the creative tasks are not solvable even by human. To address this, our MCU benchmark can create executable configurations for open-ended tasks, and ensure intra-task and inter-task diversity to simulate real game playing in different difficulty levels, while preserving solvability of tasks.

## 3.2 AUTOMATIC TASK GENERATION

### 3.2.1 ATOMIC TASK

As demonstrated above, diversity is a crucial characteristic of effective benchmarks. Intuitively, this suggests that more tasks should be included. However, if tasks consistently overlap in skill assessment (e.g., mine stone with a wooden pickaxe, mine stone with a stone pickaxe, and mine stone with a golden pickaxe Fan et al. (2022)), they merely test the same fundamental skill with minor variations. This leads to an artificial inflation of task quantity without contributing meaningfully to the evaluation of generalization. In our work, we introduce the concept of an *atomic task*, which is characterized by distinct challenges aimed at promoting genuine generalization. An atomic task is defined by two core properties:

**Goal-oriented definition**. An atomic task $\mathcal{T}$ is a basic unit defined exclusively by its goal $g$, independent of the methods, tools, or specific environmental conditions. During evaluation, the atomic task is instantiated, which induces a task-specific initial state distribution $\mathcal{P}(s_0|g)$ (see 3.2.2). For example, the atomic task "mine stone" is goal-centric, and across different evaluation batches, it may be instantiated into different $s_0$ states, such as "mine stone with a wooden pickaxe" or "mine stone with a stone pickaxe on the rainy day." However, all of these instances correspond to the same atomic task, ensuring the independence between different atomic tasks .

**Composability**. Atomic tasks can be combined to form more complex tasks by using logical operators such as "and" ($\bigwedge$) and "or" ($\bigvee$), or by introducing constraints like "when," "where," and "how." For instance, an agent could be tasked to "[mine oak log] or [mine grass] bare-handed and then [craft sticks]," where "[]" denotes individual atomic tasks. This compositional approach enables a vast task space to be explored, leveraging the combinatorial complexity of atomic goals.

The above two properties enable us to generate endless distinctive tasks. We collect over 3,000 *atomic tasks*[3] that represent unique functionalities in Minecraft. These tasks span a wide spectrum in Minecraft domain and can compose almost all the feasible tasks for junior human players. The annotated atomic tasks will also be released to community and researchers can DIY their open-ended tasks using atomic tasks as building blocks freely.

---

[3]The set is still growing.

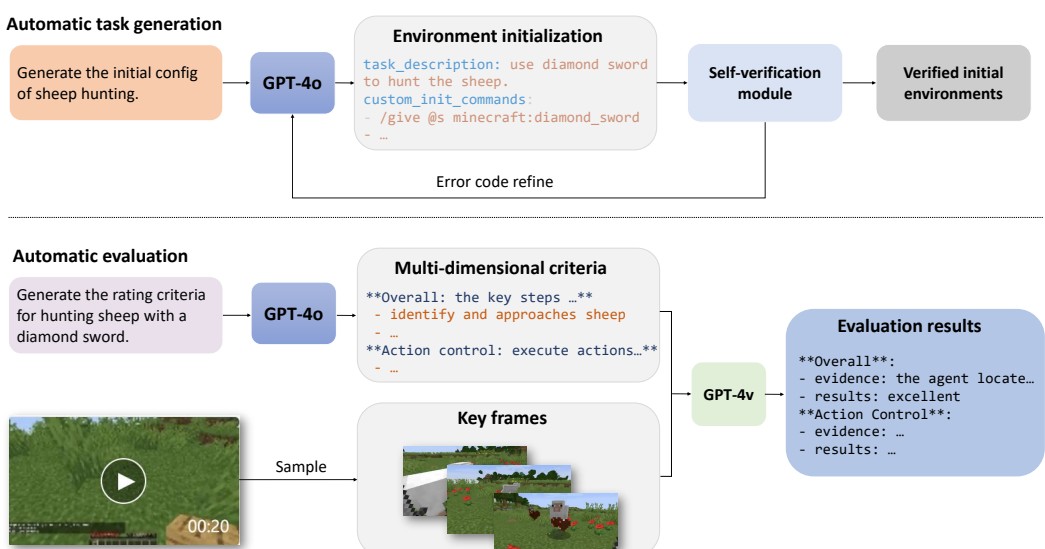

Figure 3: Automatic pipeline of task generation and performance evaluation.

### 3.2.2 LLM-POWERED SCENE GENERATION

For a given atomic task $\mathcal{T}$, we automatically generate a task-specific initial state distribution $\mathcal{P}(s_0|\mathcal{T})$ that meets two key criteria:

- The initial state distribution exhibits high diversity.
- They need to be formulated into executable files within Minecraft.

To achieve above objectives, we propose a scalable task generation pipeline powered by LLMs, coupled with an automatic verification process to ensure accuracy. LLMs provide scalability beyond human-written programs, generating a broader spectrum of task scenarios by integrating broad knowledge and creativity.

As illustrated in Figure 3, we input the atomic task and few-shot examples into GPT-4O, expecting it to generate a specific task description (used as instructions for LLM-based agents (Lifshitz et al., 2023) and for generating evaluation criteria 3.3), along with a set of executable "cheat commands" that initialize the environment configuration. This initial configuration includes various attributes such as the spawn point, inventory, equipment, items in the agent's main hand and off-hand, nearby entities, time of day, weather conditions, and more. For example, if the atomic task is to mine diamonds, the initial environment would include nearby diamond ore and an iron pickaxe in the agent's inventory. To increase task diversity, we introduce random additional conditions related to the task (e.g., placing other ores nearby) and shuffle item arrangements to prevent predictable patterns.

To address common errors generated by LLMs, we integrate soft constraints into the prompts. LLMs often struggle with numerical accuracy and game-specific rules. To mitigate this, we implement constraints to guide the outputs. For example, when generating scenes for crafting, where exact materials are required (e.g., three wool blocks and three wooden planks), we instruct LLMs to generate a surplus to account for their insensitivity to quantities. Constraints also prevent the generation of inaccessible structures (e.g., via the /fill command), maintaining environmental integrity. Essential elements like crafting tables and furnaces are consistently reminded to ensure usability.

### 3.2.3 AUTOMATIC VERIFICATION PIPELINE

To ensure the quality of generated task, Mineflayer (PrismarineJS, 2024) is employed as a super-agent to conduct task verification. We validate generated scenes by executing tasks $\mathcal{T}$ within an initial environment $s_0$ for a maximum duration $d = 60$ seconds. Let $\mathcal{V}(g, s_0, d)$ represent the task execution process. If the agent successfully completes the task within the time limit, the scene is validated. If $\mathcal{V}(g, s_0, d)$ results in failure (i.e., the task is not completed within $d$), an error signal $\epsilon$ is sent back

to the LLM. This feedback, denoted as $\mathcal{F}(\epsilon)$, prompts the LLM to generate a revised scene $s_0'$. The process ensure that the generated scenes meet benchmark standards for accuracy and usability.

### 3.3 AUTOMATIC EVALUATION

In open-world scenarios, traditional benchmarks often fall short due to the diverse and open-ended nature of tasks. In this section, We introduce an automated evaluation method designed to scale task assessments beyond the limitations of human judgment. Our framework consists of two main components. (1) Criteria generation: establishing clear, task-specific evaluation dimensions. (2) Scoring based on criteria: using these predefined dimensions to infer "scoring points" from videos of agent performance (see Figure 3).

**Criteria generation**   We define six key dimensions for evaluating agent performance:

- **Task progress:** measures critical steps and factors required for task completion.
- **Action control:** evaluates the agent's ability to avoid unrelated or unnecessary actions.
- **Material usage:** evaluates the ability in the selection and application of materials.
- **Task efficiency:** focuses on minimizing unnecessary repetitions and optimizing strategies.
- **Error recognition:** assesses the agent's capacity to identify and correct its own errors.
- **Creative attempts:** recognizes innovative approaches taken by the agent in task execution.

The LLMs can autonomously generate tailored criteria for each task. This dynamic approach allows for efficient, task-specific evaluation standards across a wide variety of tasks. These six metrics provide a comprehensive view of the agent's capabilities, offering insights into both strategic execution and adaptive problem-solving.

**Scoring with criteria**   Given the task $\mathcal{T}$ and initial states $\mathcal{P}(s_0|g)$, an agent $\mathcal{A}$ will rollout the trajectories based on its policy $\mathcal{A} : (s_0; g) \mapsto (a_0, s_1, a_1, \cdots, a_t, s_t)$, where $\{s_i\}_{i=0}^t$ are past and current states, and $\{a_j\}_{j=0}^t$ are past and current actions. We store the agent's rollout trajectories in video format. In the evaluation phase, we leverage the VLM to analyze agent performance. To optimize resource utilization, we extract one frame from every $n$ frames of the video. While this sampling approach may result in a certain degree of performance loss, it is possible to achieve a trade-off between resource conservation and evaluative efficacy hat aligns with researcher's specific conditions.

We input the sampled frames and task-specific criteria into VLM. To ensure rigor, VLM provides evidence and explanations before assigning a score (**?**). It evaluates each dimension by identifying supporting evidence from the video to justify the rating. We define the scoring intervals for each criterion as follows: *very poor, poor, fair, good, and excellent.* This structured scoring scale helps the VLM intuitively interpret performance levels, promoting consistent and detailed assessments that lead to more instructive resuplts.

## 4 EXPERIMENTS

To show that our MCU is implementable in real evaluation practice, we first validate the rationality of the automatic evaluation methods by comparing their judgments with human assessments. Subsequently, to investigate the capabilities of the existing agents, we conduct experiments in accordance with the task design principles outlined in Section 3.

### 4.1 AUTOMATIC EVALUATION

We implemented two distinct evaluation methods: comparative assessment and individual rating.

- Comparative assessment: it allows for direct comparison between two videos.
- Single rating: it scores individual video, quantifying the overall skill set of the agent.

These two approaches each have their own utility. Comparative assessment can facilitate the evaluation of an agent's improvement across different training iterations or enables the comparison between different agents combined with an Elo rating system. Individual rating provides a clear and intuitive representation of the agent's performance, allowing for the identification of specific strengths and areas for improvement.

Our video sets consisted of 60 tasks, featuring over 500 trajectories from both agent simulations and human gameplay videos. This presents a challenge for automated evaluation methods. Unlike the majority of previous work, which typically contrasted successful and unsuccessful trajectories, our dataset predominantly consists of trajectories from similar agents across different rollouts. These trajectories exhibit highly similar poses for many steps, thereby increasing the evaluative complexity. We hire 20 experts in the field of Minecraft to annotate data, with each person contributing one hour of annotation work.

**Comparative assessment**  We randomly sample two videos from the same task for each evaluation instance. Participants are then prompted to vote on the comparative quality of the videos, with options ranging from "a is better," "b is better," "tie," to "both are bad." This methodology allows for the pairing of any videos that complete the same task, creating an extensive sample space for analysis. Automated evaluation metric exhibits strong concordance with human assessments across all dimensions (Table 3). Our methodology demonstrates a marked improvement over $MineClip$, which finetune on large-scale Minecraft videos based on $CLIP_{openai}$ model (Table 2).

Table 2: The automatic evaluation results align with human judgments across a variety of tasks. Numbers represent the F1 scores for classifying the better trajectory.

| Model | Survive | Build | Craft | Tool | Collect | Explore | Average |
|---|---|---|---|---|---|---|---|
| MineClip (Fan et al., 2022) | 11.0 | 45.0 | 44.0 | 44.0 | 73.0 | 0.0 | 44.0 |
| Ours (w/o criteria) | **100.0** | 73.0 | 53.0 | **100.0** | 49.0 | **100.0** | 73.0 |
| Ours (w criteria) | **100.0** | 85.0 | 62.0 | 58.0 | 73.0 | **100.0** | **80.0** |

Table 3: The automatic evaluation results align with human judgments across different dimensions.

| Metric | Task Progress | Action | Error Recog. | Creative | Efficiency | Material | Average |
|---|---|---|---|---|---|---|---|
| F1 Score | 80.0 | 96.0 | 86.0 | 100.0 | 92.0 | 91.0 | **90.8** |

**Single rating**  In an experiment spanning five independent rating scales, the concordance between VLM and human assessment, as indicated by Kendall's $\tau$, stands at a robust 0.78, with a P-value of $1.70 \times 10^{-15}$ (see Figure 4). Our unified rating system demonstrates reliable performance on creative tasks, including 'build' and 'find', providing meaningful insights into open-ended evaluations. However, for meticulous tasks such as 'craft', which require acute attention to detail and the recognition of minor elements, the system's efficacy is somewhat diminished. Enhancements may be achieved by increasing the frame sampling rate from the current one frame per thirty.

## 4.2 How Capable Are the Existing Agents?

To show that our MCU is implementable in real evaluation practice and investigate how capable the existing agents are, we conduct experiments following the guidance of the task design principles introduced in Section 2.

### 4.2.1 Experimental settings

**Minecraft Agents.** We compare four powerful agents in Minecraft, which have been pre-trained on large-scale Minecraft video datasets to ensure generalizability: (1) VPT(bc), which is a behavior cloning model fine-tuned from *earlygame_keyword* data of YouTube video pre-training(VPT) (Baker et al., 2022); (2) VPT(rl), which is a RL fine-tuned model based on earlygame_keyword to maximizing the reward of obtaining diamond in Minecraft; (3) STEVE-I (text) (Lifshitz et al., 2023), which follows text instructions to solve tasks; and (4) GROOT (Cai et al., 2023b), which solves a task by watching a reference video. More model details can be found in Cai et al. (2023b).

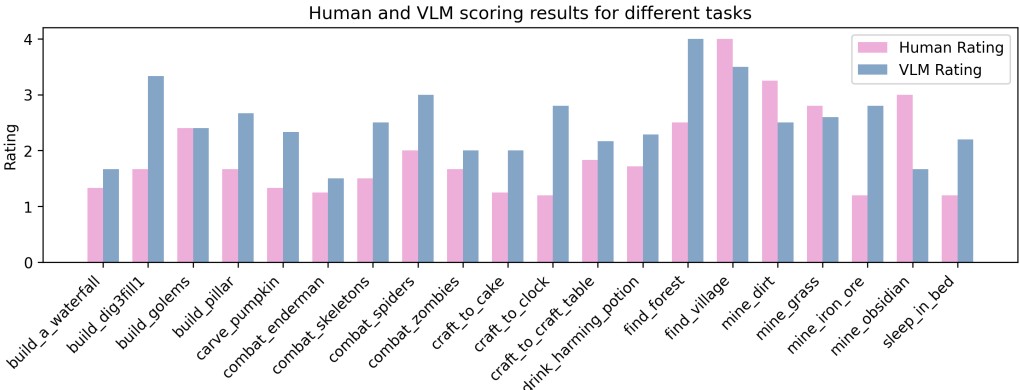

Figure 4: Human and VLM scores for various task variants demonstrate a consistent trend. When VLM scores, it extracts one image per every 30 frames, which may lead to a certain degree of information loss.

**Task Settings.** To verify intra-task diversity and inter-task diversity proposed in Section 2, we select a diverse range of tasks and establish a gradient of difficulty levels ranging from simple to hard within each task. We randomly choose 30 atomic tasks and 5 diverse compositional tasks to evaluate the agents capability. The representative tasks include "drink hurting potion", which is very novel as players rarely do this in Minecraft because it will hurt themselves, and "prepare a birthday present", which is not pre-defined in Minecraft and highly creative. Moreover, we provide three settings of difficulty level: simple, medium, and hard. The higher the difficulty level, the greater the number of factors that can impede the completion of tasks.

### 4.2.2 INTER-TASK GENERALIZATION

For ease of presentation, we categorize tested tasks into six major categories, which include many sub-categories, such as tool-use with sub-tasks like drink, carve, compose, etc., each assessing different types of skills (Figure 5). We test each task at three levels of difficulty, with 10 rollouts for each, and averaged their success rates. While agents show satisfactory performance on specific tasks like "find forest" and "mine grass," giving an illusion of impressive inter-task generalization, their performance deteriorates when faced with a broader spectrum of challenges, particularly in areas such as "craft" and "build." Notably, there is a consistent failure among all agents to execute tasks involving structured construction, exemplified by the "build nether portal" task. Furthermore, tasks requiring extensive knowledge and meticulous operational control, such as "compose obsidian," pose considerable difficulties. These results underscore the need for progress in spatial understanding and fine motor control as we advance towards the development of a generalist agent.

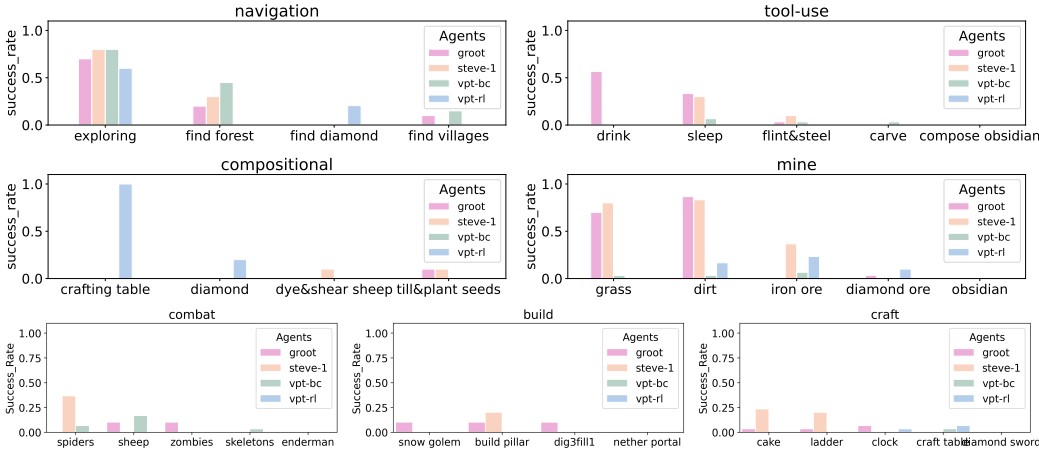

Figure 5: Testing performance across a multitude of task types, averaging from 3 difficulty levels, with 10 trial runs for each task. Video instructions are offered to Groot, text instructions are offered to Steve-1, and no instructions are offered to VPT.

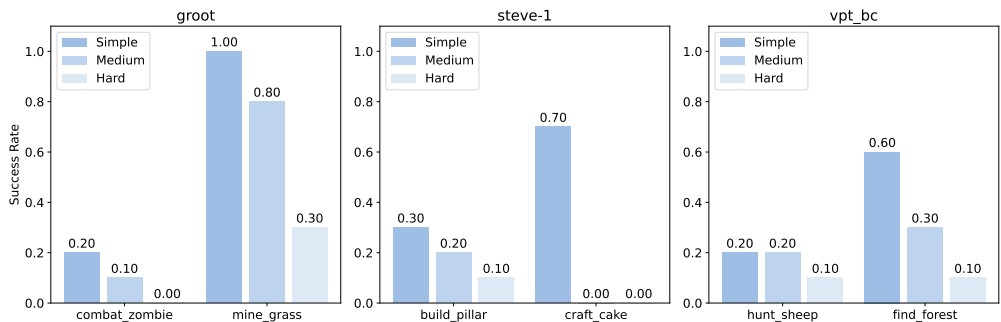

Figure 6: Generalization performance from 'simple' to 'hard' level. Results averaged from 10 trails.

We believe this vast performance gap between different level is worth highlighting. It reveals a crucial hidden flaw in training on environments that follow a fixed mode. These results underscores the necessity for developing not just basic competence in straightforward scenarios, but also the advanced resilience and discernment essential for successfully navigating the intricate and distracting challenges presented by more complex environments.

Table 4: Average performance across all tasks in different dimensions.

| Metric | Task Progress | Action | Error Recog. | Creative | Efficiency | Material | Average |
|--------|--------------|--------|--------------|----------|------------|----------|---------|
| **Vpt-rl** | 34.61 | 31.50 | 10.31 | 3.62 | 23.43 | 28.25 | 21.97 |
| **Vpt-bc** | 34.45 | 29.69 | 9.65 | 6.35 | 19.38 | 38.02 | 22.26 |
| **Steve-1** | 41.84 | 38.84 | 15.26 | 7.90 | 24.40 | 38.15 | 27.73 |
| **Groot** | 48.39 | 42.77 | 16.23 | 9.58 | 31.71 | 46.25 | **32.99** |

The averaged performance of Groot, steve-1 and VPT model across all tasks shows in Table 4. It can be observed that the Groot model performs the best, with its ranking consistent with that of humans elo rating Figure 10. However, all models show poor performance in error recognition and creativity dimensions. This indicates that there is still significant room for improvement in these aspects for the agents.

### 4.2.3 INTRA-TASK GENERALIZATION

We randomly selected two tasks where each agent performed well under the "simple" setting and investigated their performance under "medium" and "hard" difficulties. In our observations, the performance of the agents shows a significant decline as the difficulty increases (Figure 6), indicating that their generalization and robustness to interference are currently inadequate.

Taking "craft cake" as an example, Steve-1 exhibits remarkable proficiency in the simple mode, where the crafting table is readily available in hand. However, this proficiency does not scale well with increased difficulty levels. In the medium mode, where the crafting table in the inventory, and the hard mode, where additional items are present in hand, Steve-1 struggles to maintain focused execution, and becoming distracted by irrelevant information and displaying a lack of robust judgment. For agents that receive video instruction, such as Groot, relies heavily on instruction videos in many scenarios. For instance, during a test to "mine grass" where the grass is actually at its feet, but the instructional video shows the grass in front, Groot will still move to the front and perform the mining action as if that is where the target is located.

In Table 5, we can observe varying degrees of decline across multiple dimensions, but there is an increase in material usage. Analysis indicates that in the hard mode, the redundancy of items has led to an increase in the agent's usage and exploration of different tools, consequently resulting in a rise in the scores.

Table 5: Performance changes across multiple dimensions in simple and hard modes.

| Task | Task Progress | | | Action Control | | | Efficiency | | | Material Usage | | |
|---|---|---|---|---|---|---|---|---|---|---|---|---|
| | Simple | Hard | Δ | Simple | Hard | Δ | Simple | Hard | Δ | Simple | Hard | Δ |
| **enchant sword** | 62.50 | 60.00 | -2.50 | 31.25 | 30.00 | -1.25 | 18.75 | 17.00 | -1.75 | 25.00 | 50.00 | 25.00 |
| **build portal** | 81.25 | 50.00 | -31.25 | 50.00 | 40.00 | -10.00 | 43.75 | 40.00 | -3.75 | 43.75 | 60.00 | 16.25 |
| **mine iron ore** | 56.25 | 60.00 | 3.75 | 43.75 | 55.00 | 11.25 | 31.25 | 45.00 | 13.75 | 62.50 | 70.00 | 7.50 |
| **craft to cake** | 37.50 | 35.00 | -2.50 | 31.25 | 25.00 | -6.25 | 25.00 | 20.00 | -5.00 | 37.50 | 25.00 | -12.50 |
| **carve pumpkin** | 35.00 | 20.00 | -15.00 | 35.00 | 25.00 | -10.00 | 15.00 | 10.00 | -5.00 | 40.00 | 30.00 | -10.00 |
| **combat skeleton** | 25.00 | 20.00 | -5.00 | 25.00 | 20.00 | -5.00 | 16.67 | 10.00 | -6.67 | 25.00 | 15.00 | -10.00 |
| **mine dirt** | 50.00 | 65.00 | 15.00 | 40.00 | 40.00 | 0.00 | 20.00 | 25.00 | 5.00 | 40.00 | 20.00 | -20.00 |
| **sleep in bed** | 85.00 | 50.00 | -35.00 | 40.00 | 60.00 | 20.00 | 40.00 | 45.00 | 5.00 | 45.00 | 60.00 | 15.00 |
| **build dig3fill1** | 55.00 | 62.50 | 7.50 | 55.00 | 43.75 | -11.25 | 40.00 | 37.50 | -2.50 | 60.00 | 56.25 | -3.75 |
| **average** | 55.00 | 47.94 | -7.06 | 39.03 | 37.08 | -1.94 | 27.43 | 28.61 | -0.34 | 42.08 | 43.89 | 1.81 |

## 5 RELATED WORK

**Minecraft as Test Bed** Various test beds exist for multimodal generalist agents, such as Alf-World (Shridhar et al., 2020) and BabyAI (Chevalier-Boisvert et al., 2018). However, Minecraft, due to its openness and high degree of freedom, serves as a crucial platform for testing generalist agents on infinite tasks, leading to the emergence of specific benchmarks. MineDojo (Fan et al., 2022) introduced a suite of 1560 creative tasks defined by natural language instructions, but it suffers from significant redundancy and overly complex tasks that challenge practical evaluation (Lin et al., 2023). BEDD (Milani et al., 2023) presents five tasks that cover different Minecraft aspects, primarily aimed at the MineRL BASALT competition (Shah et al., 2021). By decomposing the evaluation framework, BEDD enables detailed assessments of agent performance across subgoals and characteristics like human likeness.

**Efforts to Generalist** Many agents have been developed to interact with Minecraft environments (Baker et al., 2022; Wang et al., 2023d;a; Cai et al., 2023b). Some focus on short-term task execution; for instance, Baker et al. (2022) employs imitation learning from YouTube videos, enhanced by reinforcement learning for specific tasks, but it is not a multi-task agent. Lifshitz et al. (2023) utilizes pretrained VPT and the vision-language model MineCLIP (Fan et al., 2022) to follow human instructions. These agents typically leverage pre-trained large language models (LLMs), like GPT-4 (Achiam et al., 2023) or ChatGPT (Ouyang et al., 2022), to generate action plans and execute tasks via existing low-level controllers (Wang et al., 2023d; Zhu et al., 2023b; Wang et al., 2023c;a; Ding et al., 2023). However, current LLMs, especially open-source models like LLaMA (Touvron et al., 2023), often lack the necessary knowledge of the Minecraft environment, highlighting the importance of enhancing their knowledge base for the development of generalist agents.

**LLM-as-Judge** Large Language Models (LLMs) (Achiam et al., 2023; Wang et al., 2023b) have been explored as cost-effective alternatives to human evaluation. While LLMs exhibit certain biases, such as position bias and verbosity bias (Shi et al., 2023; Zheng et al., 2023), recent advancements have mitigated these issues through techniques like providing few-shot examples to calibrate the models' scoring mechanisms (Kim et al., 2023; Li et al., 2023). Recently, state-of-the-art models have demonstrated high agreement rates with human evaluators (Liu et al., 2023), underscoring their potential to replicate human judgment in complex scenarios. The scalability and cost-efficiency offered by LLM-based evaluation address critical challenges in open-world domains (Stanley et al., 2017; Standish, 2003), providing a promising direction for future research and application.

## 6 CONCLUSION

In this work, we present the MCU framework, an automated benchmarking methodology that integrates task generation, verification, and evaluation. With evaluation results achieving an agreement rate exceeding 90%, it becomes possible to conduct large-scale assessments of diverse tasks. Moreover, MCU reveals critical limitations in the generalization capabilities of current agents, highlighting the urgent need for more comprehensive and rigorous benchmarks. We anticipate that MCU will contribute to the advancement of more versatile and truly generalist agents, empowering the research community to expand the frontiers of agent generalization.

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

## A   APPENDIX

## B   MINECRAFT ENVIRONMENT SETTING

In the regular Minecraft game, the server (or "world") always runs at 20Hz while the client's rendering speed can typically reach 60-100Hz. To ensure consistency with the server, the frame rate is fixed at 20 fps for the client. The action and observation spaces in our environment are identical to what a human player can operate and observe on their device when playing the game. These details will be further explained in subsequent subsections. Additionally, diagnostic information such as in-game stats, contents of the agent's inventory, and whether any in-game GUI is open is provided by the environment. This information can only be used for tracking, recording, and evaluating purposes but cannot serve as inputs to evaluated agents.

### B.1   MINECRAFT GAME WORLD SETTING

We have chosen to conduct the test in Minecraft version 1.16.5's survival mode. During this open-world experiment, the agent may encounter situations that result in its death, such as being burned by lava or a campfire, getting killed by hostile mobs, or falling from great heights. When this happens, the agent will lose all its items and respawn at a random location near its initial spawn point within the same Minecraft world or at the last spot it attempted to sleep. Importantly, even after dying, the agent retains knowledge of its previous deaths and can adjust its actions accordingly since there is no masking of policy state upon respawn.

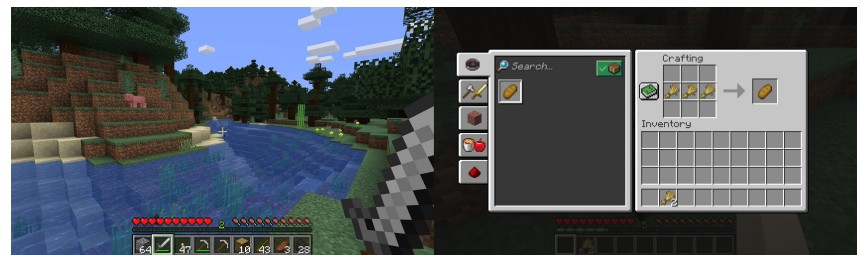

Figure 7: Minecraft game observation.

### B.2   OBSERVATION SPACE

The observation space for a human player is limited to the raw pixels visible on the display screen. It does not include any hidden information from the game world, such as hidden blocks or nearby mobs. Additionally, any information contained in the pixels must be perceived by the model rather than directly given, including inventories and health indicators. Human players can access this information by pressing F3, which should be considered part of the game screen. There are no restrictions on optional parameters that human players can adjust in the display settings, such as field of view, GUI scale (controlling the size of in-game GUI), and brightness. The rendering resolution of Minecraft is 640x360; however, it is recommended to resize images to lower resolutions for better discernibility and computational efficiency.

### B.3   ACTION SPACE

The action space is also consistent with human-playing settings, i.e., mouse and keyboard controls. These actions include key presses, mouse movements, and clicks. The specific binary actions that are triggered by keypress are shown in Table Table 6. In addition to actions triggered by keypresses, the action space also includes mouse movements. Similar to human gameplay, when there are no in-game GUIs open, moving the mouse along the X and Y axes changes the agent's yaw and pitch respectively. However, when a GUI is open, camera actions shift the position of the mouse cursor. The mouse movements are relative and adjust their position or camera angle based on their current state.

Table 6: Binary actions included in the action space. More details can be found at Minecraft wiki page[5].

| Action | Human action | Description |
|---|---|---|
| forward | W key | Move forward. |
| back | S key | Move backward. |
| left | A key | Strafe left. |
| right | D key | Strafe right. |
| jump | space key | Jump. |
| inventory | E key | Open or close inventory and the 2x2 crafting grid. |
| sneak | shift key | Move carefully in the current direction of motion. In the GUI it acts as a modifier key: when used with an attack it moves item from/to the inventory to/from the Hotbar, and when used with craft it crafts the maximum number of items possible instead of just 1. |
| sprint | ctrl key | Move fast in the current direction of motion. |
| attack | left mouse button | Attack; In GUI, pick up the stack of items or place the stack of items in a GUI cell; when used as a double click (attack - no attack - attack sequence), collect all items of the same kind present in inventory as a single stack. |
| use | right mouse button | Place the item currently held or use the block the player is looking at. In GUI, pick up the stack of items or place a single item from a stack held by the mouse. |
| drop | Q key | Drop a single item from the stack of items the player is currently holding. If the player presses ctrl-Q then it drops the entire stack. In the GUI, the same thing happens except for the item the mouse is hovering over. |
| hotbar.[1-9] | keys 1 – 9 | Switch active item to the one in a given hotbar cell. |
| show debug screen | F3 key | See the chunk cache, the memory usage, various parameters, the player's map coordinates, and a graph that measures the game's current frame rate. |

# C   WHY MINECRAFT IS SUITABLE FOR GENERALIST AGENT?

## C.1   COMPLEXITY

The environment in Minecraft is highly complex, encompassing various elements such as blocks, creatures, terrain, vegetation, and more. This complexity poses diverse challenges for agents in this environment, requiring them to learn to adapt and address a wide array of intricate tasks. The intricate nature of the environment provides generalist agents with abundant learning opportunities, enabling them to flexibly navigate through different scenarios.

## C.2   OPEN-ENDEDNESS

Minecraft offers a vast open world where players can freely explore various regions. This openness exposes agents to limitless potential environments, requiring them to possess exploration and navigation capabilities. In an open world, a generalist agent must be able to adapt to new terrains, scenes, and situations. In this open-ended environment, it becomes more convenient to select tasks with varying difficulty levels, each presenting unique dimensions of challenge. This allows us to evaluate the agent's performance in a targeted manner, assessing its proficiency across various abilities.

## C.3   DYNAMISM AND UNPREDICTABILITY

Compared to some static, text-based, or vision-language based test environments, the dynamism and unpredictability of Minecraft provide unique advantages for the training and testing of intelligent agents. The in-game environment is filled with dynamic changes and unknown factors, including day-night cycles, the random appearance of creatures, diverse terrains, and more. This dynamism and unpredictability necessitate that the agents adapt flexibly to various scenarios and possess the ability to handle unexpected events, thereby better cultivating their generalization skills for complex environments.

## C.4 CREATIVITY AND INNOVATION

Minecraft allows for a high degree of creativity and innovation. The abundance of open-ended tasks, such as construction and decoration, provides agents with ample space to unleash their creativity. By exploring various ways to achieve goals, agents cultivate innovative abilities in addressing diverse and complex challenges.

## C.5 BROAD CHALLENGE COVERAGE

Minecraft, with its outstanding freedom and depth of tasks in an open-world setting, is well-suited as a training and testing platform for a generalist agent. In general, agents in the Minecraft environment may encounter the following four primary challenges.

**Long-horizon Decision Making** In the Minecraft environment, tasks can break into a sequence of subtasks. For instance, to achieve the goal of mining diamonds, players need to complete various subtasks including chopping the trees, crafting a crafting table, obtaining a pickaxe, and searching for diamonds. The sequences of subtasks for a task are not necessarily the same; rather, they can be entirely different. For example, to complete the task of "obtaining wool," the typical approach is to kill sheep. However, if there are no sheep nearby, players might need to kill spiders to obtain string, then use the string to craft wool, or even directly trade with villagers for wool. Different environments lead to different subtask sequences, placing a challenge on the agent's ability for long-horizon decision-making.

In this context, agents must have the capability to predict and plan future environments and actions, instead of mere reactions to the current state. Long-horizon decision-making in Minecraft requires agents to understand the complex spatial and temporal relationships within the game environment. The diverse and dynamic nature of tasks, combined with the multitude of possible approaches, demands that agents develop a comprehensive understanding of the environment to effectively navigate through various steps and reach their goals.

**Precise Control** As a well-known sandbox construction game, Minecraft allows players to engage in intricate building and operations. Therefore, tasks related to building and crafting are important components of the Minecraft task list. This task always involves precise movement, accurate block placement, and destruction. For example, the task "building a nether portal" requires the agent to build at least a $2 \times 3$ rectangular frame with specific blocks. If the player placed the block in a wrong position, they should mine that wrong block with the pickaxe and place it again. This demands precise and accurate control. RL agents need to handle high-dimensional action spaces and achieve precise control in the environment to accomplish complex tasks. This presents a challenge for the stability and precision of the agent.

**Out-of-distribution Generalization** The Minecraft environment is dynamic, filled with various possible scenarios and conditions. The terrain, ecology, organisms, and even the weather are ever-changing, and it's impossible for the agent to encompass all of them in the training data. On the other hand, due to the fact that the vast majority of training data consists of reasonable behaviors, the model's free exploration or learning errors make it prone to encountering environments not present in the training data, such as falling into the lava when mining the diamond. How to enable the model to generalize to out-of-distribution environments and adapt to the complexity of the ever-changing open-world is a notable challenge.

**Compositional Generalization** To adapt to the long-horizon and varied tasks in Minecraft, the model needs to have the ability of compositional generalization. For example, when the training set includes data crafting sticks from planks and crafting ladders from sticks, we hope the model can generalize the ability to craft ladders from planks. The Minecraft environment offers nearly infinite combinations of tasks, with the majority of them not appearing in the training set. Accomplishing these tasks poses a significant challenge to the model's compositional generalization capability.

## C.6 COMMUNITY AND RESOURCES

The game Minecraft boasts a vast and active community where players share rich experiences, creativity, and problem-solving techniques. This communal sharing environment provides a massive resource pool for agents, enabling them to draw knowledge, inspiration, and skills from the community. By engaging with the community, agents can tap into the wisdom of diverse players, enhancing their ability to perform tasks more comprehensively and effectively in the open-world setting.

Furthermore, the Minecraft community has fostered a wealth of mods and plugins, allowing players to customize their gaming experiences. This provides agents with diverse and targeted training and testing scenarios, aiding in the development of their adaptability to different environments. The creative spirit and resource-sharing ethos within the community further enrich the learning experience for agents, enabling them to draw upon and apply information from a wide-ranging community. Therefore, as a community-driven platform, Minecraft offers abundant social and knowledge resources for the training and testing of open-world agents.

## C.7 SAFE AND CONTROLLED

Minecraft provides a safe and controlled virtual world, offering an ideal space for model learning. The safety of this environment allows models to learn in virtual reality without the potential risks associated with real-life situations. Additionally, Minecraft offers a high degree of controllability, enabling researchers to customize tasks and adjust environmental parameters to precisely manage the learning scenarios for models. This control is advantageous for the agent to learn and optimize performance in specific tasks. Therefore, as a secure and controlled virtual environment, Minecraft offers a unique and adaptable training platform for reinforcement learning models.

Table 7: Partial definition and examples for task category.

| Category | Definition | Example |
|---|---|---|
| Crafting Task | The tasks accomplished through the in-game inventory interface Typically require specific functional blocks to complete, such as crafting (requiring a crafting table), enchanting (requiring an enchanting table), potion brewing (requiring a brewing stand), smelting (requiring a furnace) and so on. Players need to accurately drag items using the mouse to their respective slots and then press the confirm key. | Craft to diamond pickaxe Enchant book Craft to baked potato Craft to awkward potion |
| Navigation Task | Navigation or movement tasks involve finding specific terrain, ecosystems, creatures, items, or other targets. | Find a zombie Find blackstone Find forest Find village |
| Mining Task | The task of breaking blocks, and extracting resources like ores, sometimes requiring specific tools such as an iron pickaxe. | Mine dirt Mine grass Mine diamond ore Mine dragon egg |
| Tool-Use Task | The tasks primarily involve using the in-game interaction key (i.e., right mouse button). to interact with items, such as eating food, planting, feeding animals, and using specific blocks like crafting tables. | Eat bread Breed a cow Interact with crafting table Light TNT |
| Building Task | Building and construction tasks involve building various shapes and structures. The final outcome of the construction may not be identical and usually allowing for a degree of openness and creativity. | Bbuild a tower Build a fence Build a Nether Portal Build a castle |
| Trapping Task | A special type of interactive task between creatures and agent, often aimed at restricting the movement of entities, such as guiding creatures, controlling their paths, breeding, etc. This may involve the use of tools such as boats, leads, fishing rod or other related items. | Trap a zombie with a boat Hook a sheep using fishing rod Bring a cow into nether Trap a creeper in house |
| Motion Task | Tasks that focused on the action of agent itself as the goal, mostly operations or skills that player used in games. | Sneak Drop an item Dive deeply MLG water bucket |
| Decoration Task | Tasks aimed at enhancing the visual appeal of the game environment, Often associated with creative aspects. | Clean the weeds Light up a cave Decorate the home Hang item |

## C.8    COMPOSITION OF ATOMIC TASK

Our atomic task list exhibits high diversity. By combining atomic tasks with "and" ($\bigwedge$) and "or"($\bigvee$) grammar, or added constraints like "when, where, how", we can generate a vast array of complex tasks in the Minecraft environment. We conducted research on tasks used in previous Minecraft work, all of which can be expressed using this method by our atomic tasks. Table 10 shows some examples of complex tasks expressed as combinations of atomic tasks.

Table 8: Examples of decomposition of tasks recently used in work in Minecraft to atomic tasks.For arbitrary task t, '[t]' means an atomic task t or the decomposition of task t to atomic tasks. Once we deduced the expression of [t], we are able to use [t] to express the decomposition of more complicated tasks, thus omitting the need for complicated and repetitive expressions.

| literature | Task | Decomposition |
|---|---|---|
| Cai et al. (2023a) | Mine oak wood | [find oak wood] and [mine oak wood] |
| | Hunt sheep | [find a sheep] and [mine oak wood] |
| | Mine dirt | [find dirt] and [mine dirt] |
| | Obtain wool | ([find a sheep] and [hunt a sheep]) or ([find a sheep] and [shear sheep]) or ([find a spider] and [craft to white wool]) |
| Lifshitz et al. (2023) | Collect seeds | ([find grass] and [mine grass]) or ([find tall grass] and [mine tall grass]) |
| | Chop a tree | ([find oak log] and [mine oak log]) or ([find spruce log] and [mine spruce log]) or ([find birch log] and [mine birch log]) or ([find jungle log] and [mine juggle log]) or ([find acacia log] and [mine acacia log]) or ([find dark oak log] and [mine dark oak log]) or ([find stripped spruce log] and [mine stripped spruce log]) or ([find striped birch log] and [mine stripped birch log]) or ([find stripped jungle log] and [mine stripped jungle log]) or ([find stripped acacia log] and [mine stripped acacia log]) or ([find stripped dark oak log] and [mine stripped dark oak log]) or ([find stripped oak log] and [mine stripped oak log]) |
| Baker et al. (2022) | Obtain crafting table | [chop tree] and [craft to planks] and [craft to crafting table] |
| | Mine diamond | [chop tree] and [obtain crafting table] and [craft to wooden pickaxe] and [find stone] and [mine stone] and [craft to stone pickaxe] and [find iron ore] and [mine iron ore] and [craft to furnace] and [find coal ore] and [mine coal ore] and [craft to iron ingot] and [craft to iron pickaxe] and [find diamond ore] and [mine diamond ore]. |

# D    DIFFICULTY SCORES

## D.1    HUMAN ANNOTATION

To get an annotation for task difficulty scores of our selected tasks in difficulty and essence, we designed and distributed a questionnaire to collect what human players who are familiar with Minecraft think about them. The questionnaire includes two parts, the quiz part and the annotation part. The quiz part contains five multiple-choice questions with 25 options to test their familiarity with Minecraft; each correctly answered option is worth 1 point. Then we filtered out the questionnaires with a correct rate of less than 75%, and then considered their investigation parts for the remaining questionnaires. The quiz is shown in Table 9. We distributed the questionnaires in the Minecraft community and collected a total of 76, with 76 of them were valid.

In the annotation part, the respondents are asked to rate each selected task in the five dimensions: time consumption, creativity, novelty, intricacy, and visual diversity. We inform the annotators that the first two points are as the name implies, novelty stands for how rare or uncommon you think in real game scene, and intricacy means the extent to which the task is considered to require precise control. We also give some examples: if a player's mouse is not sensitive enough, how much will the difficulty of this task increase? The last point, visual diversity, refers to whether or not you will see rich visual information when completing this task. We use the respondents' evaluations of these five dimensions to reflect the diversity and representativeness of the tasks we selected and to verify that our selection of these tasks to evaluate Minecraft agents is reasonable.

Table 9: The quiz in our questionnaire, is used to judge the respondents' familiarity with Minecraft. The problems are adapted from Milani et al. (2023).

| No. | Question | Options |
|---|---|---|
| 1 | A bed can | A. speed up the night.
B. change the respawn location.
C. be crafted from drops of a certain animal in the game.
D. can be crafted by a furnace, but cannot be crafted by a crafting table. |
| 2 | You can acquire EXP when | A. killing hostile mobs.
B. mining trees.
C. jumping on a coal ore block.
D. mining coal.
E. enchanting a diamond sword. |
| 3 | What mobs can deal damage to the player? | A. Skeletons.
B. Zombies.
C. Sheep.
D. Pigs.
E. Creepers.
F. Enderman. |
| 4 | What items can be eaten? | A. Apples.
B. Dirt.
C. Beef.
D. Wheat.
E. Breads.
F. Spider eyes. |
| 5 | If you mine a block with a bare hand, what kinds of block can drop the corresponding item? | A. Wooden logs.
B. Wooden planks.
C. Iron ore.
D. Coal ore. |

## D.2 ANNOTATED DIFFICULTY

The annotated difficulty scores are shown in Figure 8.

# E TASK ANNOTATION

We use stratified sampling for different task groups, making the selection of tasks for each group diverse and representative, and at the same time, focus on the different groups fairly. More precisely, for each task group $r$, our selected task $\mathcal{T}$ meets

$$\mathcal{T}|r \sim \mathcal{P}(t|r)$$

and for each task $t$,

$$S_0|t \sim \mathcal{P}(s_0|t)$$

The former represents the representativeness and diversity of tasks in each group, which has been demonstrated by the high entropy of the sampled tasks. Later we will elaborate on how to manipulate our environment configurations to try to make the distribution of $s_0$ conform to the latter formula as much as possible.

In order to compare the performance of the model output with the performance of human players, video data of human players is needed. We will also annotate the videos we recorded.

## E.1 MANIPULATIONS OF A TASK

The initial state of a task contains all the information an agent can utilize condition on the agent "plans" to do the task (not only the valid input but also what it can derive or perceive), including the observed 2D pixels of the game scene, the inventories and the coordinate (which can be perceived when pressing F3, especially the y dimension). The inventories $\mathcal{I}$ include what items are necessary for the task $\mathcal{I}_n$, otherwise, the agent won't plan to do the given task in a real game, and other random inventories $\mathcal{I}_r$. This is what we can manipulate, and we need to make these random variables as close as possible to the real distribution in the game.

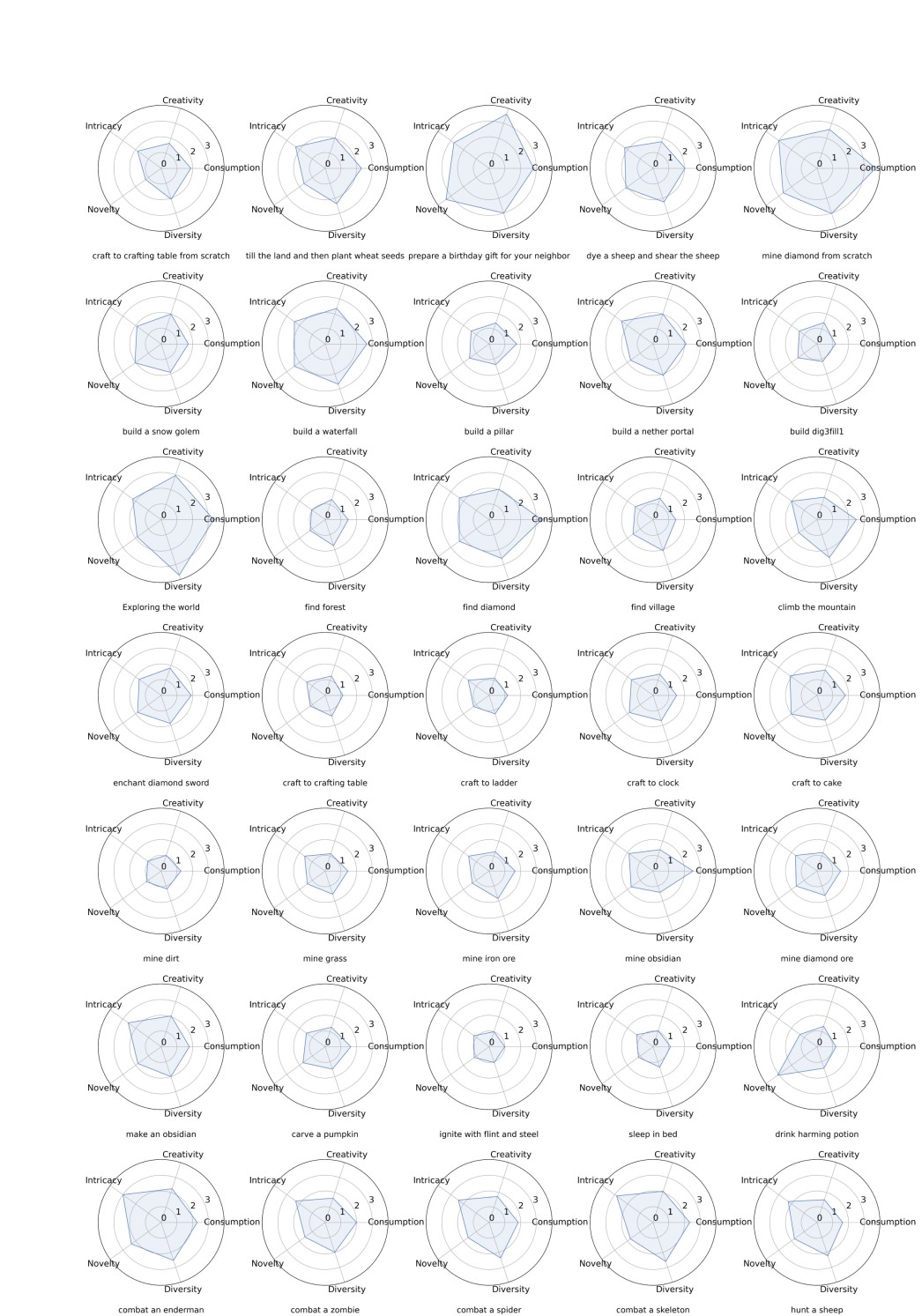

Figure 8: The annotated difficulty score for each task.

### E.1.1 OBSERVATION AND COORDINATE

For a fixed version of the Minecraft game, these two elements can be defined by the seed of the world, the coordinate, and the facing direction. The seed of the world is completely independent of other variables, so it can be selected arbitrarily. The facing direction is the same as what it was before teleported to the task scene, which is random and we do not manipulate it. If a coordinate is set to be a proper spawn location when testing a given task in a given world, it needs to meet some preconditions, which can be biome names supported within the game or other restrictions. For example, in the mission `climb the mountain`, the agent needs to spawn in `stony shore`, a kind of biome in the game, while running to a village, the location should be close to one.

We list a series of location coordinates for each selected seed corresponding to each precondition we need. Each (seed, precondition) pair can correspond to different location coordinates and can be used in different tasks. When we set up the environment configuration files, we only specify the world seed and preconditions, and when the world is loaded and generated, the location will be randomly selected from this list that meets the precondition.

### E.1.2 INVENTORIES

The inventories $\mathcal{I}$ include what items are essential for completing task $\mathcal{I}_n$, and other random inventories $\mathcal{I}_r$ as a distractor. $\mathcal{I}_n$ is also a random variable since there are different ways to approach the same goal. For example, the agent can use an iron pickaxe or diamond pickaxe to mine a diamond ore. We looked at different ways to accomplish the same task and tried to include as many of them as possible, testing different $\mathcal{I}_n$. As for $\mathcal{I}_r$, to reduce the difficulty of some task tests, we do not set $\mathcal{I}_r$, and for other task tests, we randomly sampled initial inventories from game snapshots of contractor data of VPT.

### E.2 HUMAN VIDEOS FOR TASKS

Human videos serve two purposes - they are used as reference videos for GROOT, and they are used for comparison with the trajectory videos generated by the agent models. For each task, we choose three world seeds - 19961103, 20010501, and 12345, and for each (task, seed) pair, we manipulate what we can manipulate as described above, and have three environment configuration files. For each environment configuration file, we record a human video and use the first file of seed 19961103 for GROOT citegroot reference video.

## F PROGRAMMATIC METRICS FOR STUDIED TASKS

**Metrics**   During our evaluation, we use the scripts to record information for each video, including items that are crafted, used, broken, and mined, blocks that are mined, entities that are killed, and horizontal and vertical offset. With the scripts, some tasks can be evaluated using programmatic metrics in a fully automated manner, thus saving time and human resources. Table 10 shows examples of tasks in our experiments and their corresponding metrics. The threshold for the task 'Explore the world' is 50 units, while for the task 'Climb the mountain,' it is 20 units in seed1 and 30 units in seed2.

**TrueSkill Rating**   We also evaluate and compare the previous agents through the TrueSkill rating system. It was developed by Microsoft Research and is currently used in matchmaking and ranking services on Xbox LIVE. It takes the uncertainty of the players' ability into consideration and models the score of a player as a Gaussian distribution $\mathcal{N}(\mu, \sigma^2)$, then uses the Bayesian inference algorithm to measure a player's score, where $\mu$ is the average skill of the player and $\sigma$ is the standard deviation of a play's performance. A real skill of the player is between $\mu \pm 2\sigma$ with $95\%$ confidence. The result of TrueSkill Rating is shown in Figure 9.

## G MINECRAFT ENVIRONMENT SIMPLIFICATION IN PREVIOUS WORKS

In our evaluation mechanism, we require the agent's observation space and action space similar to a human player playing in front of a device. In other words, all the information that needs to

| Task | Metric |
|------|--------|
| Build snow golem | Success rate |
| Build pillar | Success rate to build a pillar at with least 5 blocks |
| Build dig3fill1 | Success rate |
| Build nether portal | Success rate |
| Build a waterfall | Success rate |
| Craft to ladder | Success rate |
| Craft to crafting table | Success rate |
| Craft to clock | Success rate |
| Craft to cake | Success rate |
| Enchant diamond sword | Success rate |
| Combat zombies | Success rate |
| Combat spiders | Success rate |
| Combat skeletons | Success rate |
| Combat enderman | Success rate |
| Hunt a sheep | Success rate |
| Mine grass | Success rate when the number of grass blocks and tall grass blocks mined in total exceeds the threshold. |
| Mine obsidian | Success rate |
| Mine dirt | Number |
| Mine diamond ore | Success rate |
| Mine iron ore | Success rate |
| Explore the world | Success rate when the horizontal offset is greater than the threshold. |
| Find a forest | Success rate to stay in forest for last 10s |
| Find a village | Success rate to stat to village for last 10s |
| Find diamond | Success rate |
| Climb the mountain | Success rate when the height offset is greater than the threshold. |
| Drink harming potion | Success rate |
| Carve pumpkin | Success rate |
| Make fire with flint and steel | Success rate |
| Make obsidian by water | Success rate |
| Sleep in bed | Success rate |
| Dye a sheep and then shear the sheep | Success rate |
| Mine diamond from scratch | Success rate |
| Craft to crafting table from scratch | Success rate |
| Till the land and then plant wheat seeds | Success rate |

Table 10: The programmatic metric for each task.

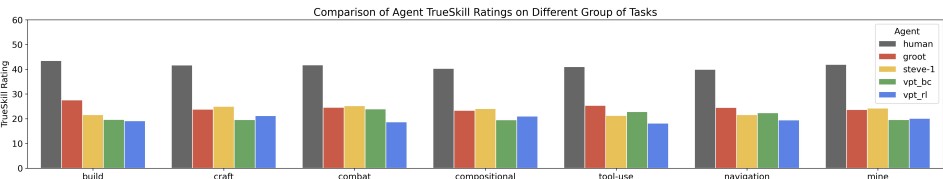

Figure 9: Comparison of Agent TrueSkill Ratings on Different Groups of Tasks

be perceived comes from the pixels displayed on the screen, and the underlying control relies on simulating mouse and keyboard operations. The only difference is that the degree of freedom is slightly lower, that is, the keyboard operations only allow the types shown in Table 6. However, in order to develop a Minecraft agent more efficiently, some previous works did not meet these requirements. Some benchmarks simplified the observation space and action space, and some previous agents further simplified the benchmarks. Some of them reduced the freedom of operation by changing the action space, others utilized some additional information within the game that cannot be obtained from the pixels.

### G.1 PREVOIUS BENCHMARKS

**MineRL** MineRL (Guss et al., 2019b) is a benchmark for Minecraft agent competition, and there are different unrelated tasks to evaluate. Before version 0.4.4, MineRL offered different action spaces and observation spaces for each task, and for each task, the spaces are exactly what is needed to

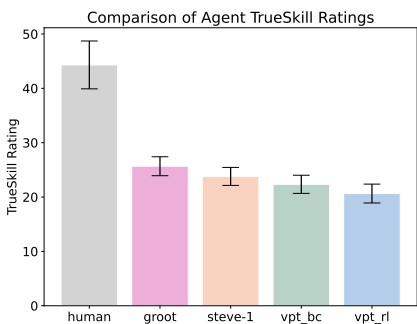

Figure 10: The TrueSkill evaluation results for the compared agents and human.

complete this track. After version 1.0.0, the observation space is the same as in this paper, and the action space is similar to ours, except for two high-level actions - "pickItem" and "swapHands".

**MineDojo**    The observation space of MineDojo (Fan et al., 2022) simplifies the environment to a large extent. Apart from the ego-centric RGB frames, it can obtain the 3D voxels, nearby tools, damage sources, and lidar, which are extra in-game information, equipment, inventory, life statistics, GPS location, and compass, which should be derived from the pixels. As for the action space, some high-level actions are encapsulated such as "craft" and "equip".

**BEDD**    The observation space of BEDD (Milani et al., 2023) is the same as ours. It requires actions to directly simulate mouse and keyboard operations but does not limit whether to encapsulate high-level actions.

G.2    PREVIOUS AGENTS

**VPT**    VPT (Baker et al., 2022) does little to simplify the environment. The only difference between VPT and our benchmark is that VPT disables F3 key, but it does not make use of the information in it.

**DEPS**    The experiment of DEPS contains two parts. Both MineRL and MineDojo benchmarks have been tested and each experiment follows the action space and observation space of the corresponding benchmark.

**Voyager**    The information used by Voyager (Wang et al., 2023a) is less similar to human players. Voyager runs in a Minecraft world by incrementally building a skill library, which stores action programs, whose code is generated by GPT-4 (Achiam et al., 2023). The observation of Voyager includes the feedback of GPT-4 and it knows its inventory directly.

**GITM**    Ghost in the Minecraft Zhu et al. (2023b) The observation space is the same as MineDojo and the action space is also structured. Some actions are very high-level, such as "explore".

**Steve-1**    Steve-1 has the same observation space and action space as VPT.

**Groot**    Groot has the almost same observation space and action space as VPT, except dropping items is not allowed (i.e., the Q key is disabled).

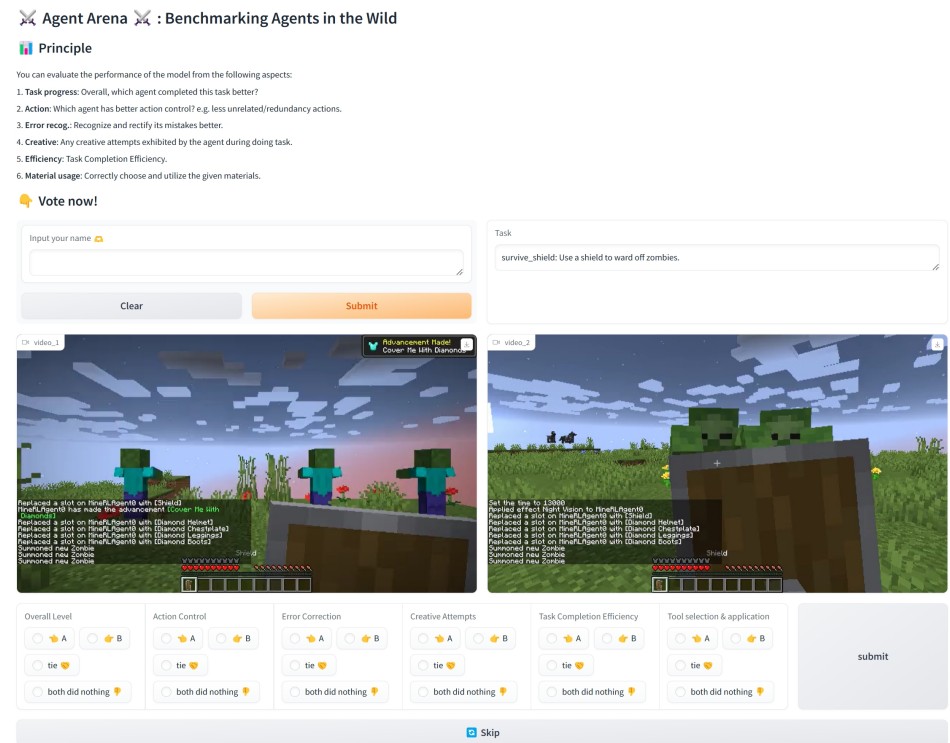

Figure 11: Video comparison website.

## G.3 HUMAN RATING SYSTEM

The human rating systems are shown in Fig11 and Fig12. Take video comparison website as an example, it is designed to evaluate agent performance by presenting two videos side by side, enabling human raters to directly compare their behaviors for the same task. The page is structured into several modules:

1. Task description module: positioned at the top-right, this module specifies the task to be evaluated (e.g., survive shield: Use a shield to ward off zombies). It ensures that raters understand the objective of the task before scoring.

2. Video display module: two videos are presented side by side. Each video provides a replay of the agents' gameplay. This visual design helps raters observe agent behaviors, mistakes, or innovative strategies in real-time.

3. Scoring panel: located below the videos, the scoring panel allows raters to assess agent performance across six dimensions.For each dimension, raters can choose which agent performed better, mark a tie, or indicate that neither agent took relevant actions.

4. Input and submission module: at the top-center, the name input box collects rater identifiers to ensure traceability. The Submit Button at the bottom sends completed ratings to the database, contributing to the dataset used for benchmarking.

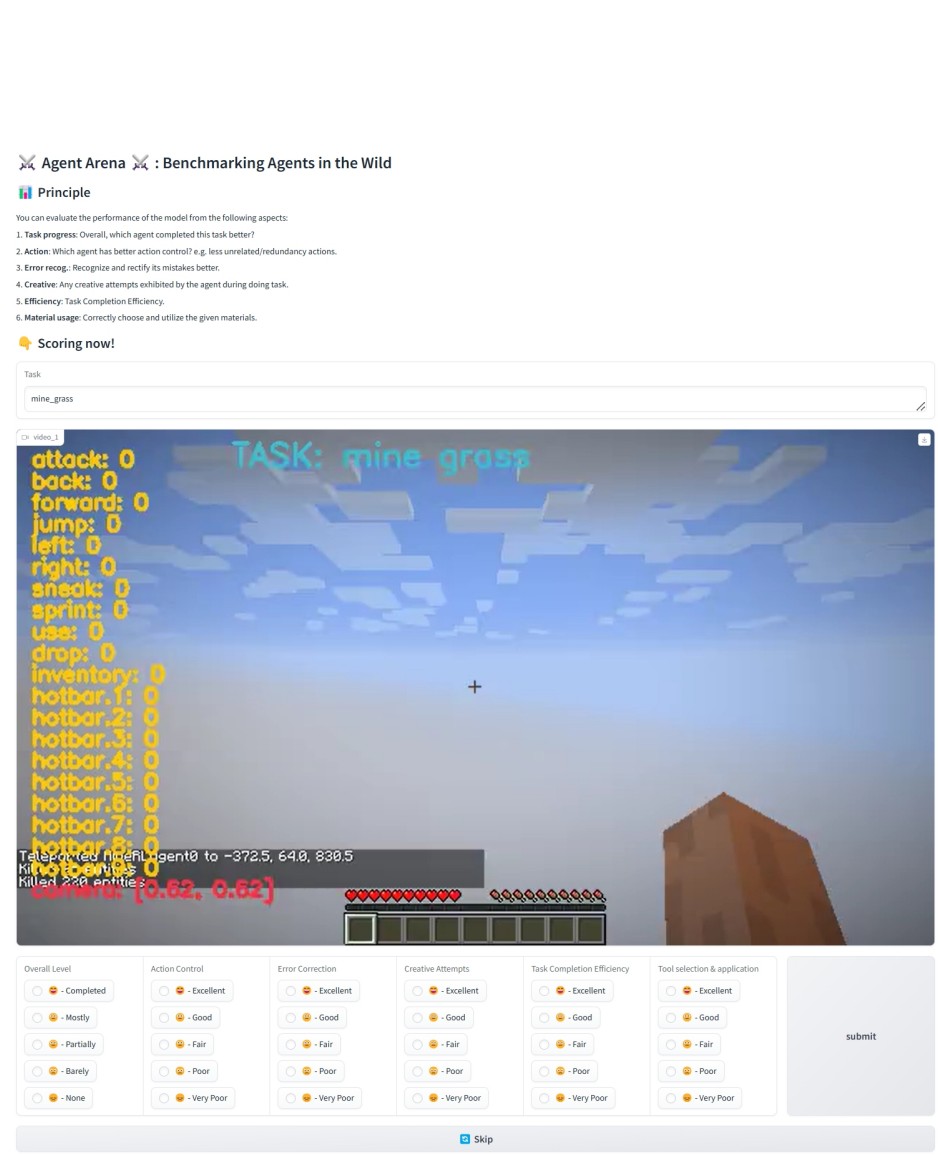

Figure 12: Individual video rating website.

### G.4 PROMPT FOR CONFIGURATION GENERATION

```
You are an expert of Minecraft, and I am a new Minecraft player.
I will give you a task name. you should generate a task description and
    give me all the necessary things I need for completing the task.

I will give you the following information:
The task I want to complete: ...
You should perform the following steps to help me:
1. Generate a description of how to do the task.
2. Tell me all valid items, mobs, biomes and all the necessary things to
    complete task;
3. Formulate step.2 information as cheat commands;
4. Randomly generate one or two related but not necessarily cheat
    commands.
5. Don't always generate the cheat commands for necessary items at the
    front and place random commands at the back. Shuffle their order.
6. Only output one sentence task description, one sentence of step.2 and
    custom\_init\_commands

e.g. The task I want to complete: Trade for iron helmet.
You should respond in the format as described as below:
- Task description: Trade with a villager to obtain an iron helmet using
    the items
you have in your inventory.
- In order to trade for iron helmet, we need at least 5 emerald and a
    armorer nearby.
- custom_init_commands:
  - /give @s minecraft:armor_stand 2
  - /give @s minecraft:emerald 10
  - /summon villager ~2 ~ ~5 {Profession:"minecraft:armorer",VillagerData
      :{profession:
  "minecraft:armorer"}}
  - /give @s minecraft:diamond 64

e.g. The task I want to complete: craft a crafting table.
You should respond in the format as described as below:
- Task description: Open inventory and craft a crafting table.
- In order to craft a crafting table, we need at least 4 planks.
- custom_init_commands:
  - /give @s minecraft:oak_planks 64
  - /give @s minecraft:bread 16
  - /time set night

e.g. The task I want to complete: mine iron_ore.
You should respond in the format as described as below:
- Task description: Find and mine the iron_ore use the right tool.
- In order to mine iron_ore, we need at least a stone pickaxe or a better
    one, and have iron_ore nearby.
- custom_init_commands:
  - /give @s minecraft:stone_pickaxe
  - /execute as @p at @s run fill ~2 ~2 ~3 ~1 ~5 ~4 coal_ore
  - /execute as @p at @s run fill ~-5 ~-2 ~-1 ~ ~ ~-3 iron_ore
  - /give @s minecraft:wooden_pickaxe

e.g. The task I want to complete: flying trident on a rainy day.
You should respond in the format as described as below:
- Task description: flying trident on a rainy day.
- In order to flying trident on a rainy day, we need a trident enchanted
    with the
riptide enchantment, and set the weather in rainy mode.
- custom_init_commands:
  - /weather rain
  - /give @p minecraft:trident
```

```
53    - /give @p minecraft:trident{Enchantments:[{id:"minecraft:riptide",lvl
         :1}]} 3
54    - /give @p minecraft:fire_charge{Enchantments:[{id:"minecraft:riptide",
         lvl:1}]} 3
55
56  Note:
57   - You should provide accurate information and executable cheat commands
         of Minecraft.
58   - The quantity of items in the cheat command should be more than what is
          required. For example, the task need at least 10 emerald, provide
         15 instead.
59   - You should provide all the tools and environments required for
         completing the task.
60   - Attention, there are certain items that cannot be directly summoned,
         such as trees, sugar cane, etc.
61   - Do not give me the final target things directly in my inventory.
62   - Some crafting tasks are not completed using the crafting table, they
         could be done with tools like the furnace, enchanting table, or
         brewing stand and so on. You need to select the appropriate tool.
63   - Remember to provide a crafting table, furnace, enchanting table,
         brewing stand or similar items, if the task requires it.
64   - When use /fill command, ensure not to generate them in inaccessible
         locations (such as high in the sky), and be extremely cautious not
         to suffocate the agent.
65   - For pick-up task, you can design the item that can be directly pick up
          by hand, like dirt or poppy.
```

Listing 1: Prompt for Configuration Generation

## G.5 PROMPT FOR VIDEO COMPARISON

```
1  You are an expert in Minecraft and excel at evaluating agents in the AI
         field.
2  I will give you a task name, a grading criterion for this task, and two
         videos (Video A and Video B) of an agent performing the task. The
         grading criterion has several major criteria (***) and several
         evaluation rules under each major criterion.
3  You need to carefully compare the agent's performance in Videos A and B
         according to the evaluation rules and output one of the following:
4  "A is better", "B is better", "tie", or "both are bad".
5
6  The more the agent complies with the rules in the criteria, the better
         their performance is.
7
8  Output "A is better" when A performed better according to the evaluation
         rules.
9  Output "B is better" when B performed better according to the evaluation
         rules.
10 Output "tie" when both videos demonstrate similar capabilities.
11 Output "both are bad" when both videos have hardly done anything related
         to the rules or have performed very poorly.
12
13 Before output the decisions, you should list the relevant evidence from
         videos to support your decisions (within 80 words), do not simply
         copy the phrases from the rules.
14 Please make the decision across six major criteria, including task
         progress, material selection and usage, action control, error
         recognition and correction, creative attempts, and task completion
         efficiency.
15
16 You should follow the following output format to organize your output.
         xxx is the placeholder. Evidence can be more than one. The output
         format should be as follows:
17    Task Progress:
```

```
18      - evidence xxx
19      result: xxx
20
21      Action Control:
22      - evidence xxx
23      result: xxx
24
25      Error Recognition and Correction:
26      - evidence xxx
27      result: xxx
28
29      Creative Attempts:
30      - evidence xxx
31      result: xxx
32
33      Task Completion Efficiency:
34      - evidence xxx
35      result: xxx
36
37      Material Selection and Usage:
38      - evidence xxx
39      result: xxx
40
41      Overall results:
42      - Task Progress: xxx
43      - Action Control: xxx
44      - Error Recognition and Correction: xxx
45      - Creative Attempts: xxx
46      - Task Completion Efficiency: xxx
47      - Material Selection and Usage: xxx
48
49  Notes:
50
51  If the evaluation rules include "e.g.", it is only an example and you
        should not be limited to the listed "e.g.". All phenomena that
        conform to the major criteria should be considered.

52
53  Task progress considers only the completion of key steps of the task and
        is unrelated to artistic qualities or similar aspects.
```

Listing 2: Prompt for Video Comparison

## G.6 PROMPT FOR INDIVIDUAL VIDEO RATING

```
1   You are an expert in Minecraft and excel at evaluating agents in the AI
        field.
2   I will give you a task name, a grading criterion for this task, and a
        video of an agent performing the task.
3
4   The grading criterion has several major criteria (***) and several
        evaluation rules under each major criterion.
5   You need to score the agent's operations in the video based on the
        evaluation rules. The more the agent complies with the rules in the
        criteria, the higher the score it receives.
6
7   - If you think the agent's behavior does not relate to the stated rule,
        score None.
8   - If you think the agent's behavior barely relates to the stated rule,
        score Barely.
9   - If the agent's behavior is partially related to the rules, score
        Partially.
10  - If the agent's behavior is mostly related to the rules, score Mostly.
11  - If the agent's behavior is completely related to the rules, score
        Completed.
```

```
12
13   If you believe the agent complies with the rule, you should list the
         relevant evidence from the video (within 50 words). Do not simply
         copy the phrases from the rules.
14   Assign an appropriate score six major criteria, including task progress,
         material selection and usage, action control, error recognition and
         correction, creative attempts, and task completion efficiency.
15
16   The output format should be as follows:
17
18       Task Progress:
19       - evidence xxx
20       Score: xxx
21
22       Action Control:
23       - evidence xxx
24       Score: xxx
25
26       Error Recognition and Correction:
27       - evidence xxx
28       Score: xxx
29
30       Creative Attempts:
31       - evidence xxx
32       Score: xxx
33
34       Task Completion Efficiency:
35       - evidence xxx
36       Score: xxx
37
38       Material Selection and Usage:
39       - evidence xxx
40       Score: xxx
41
42       Overall Scores:
43       - Task Progress: xxx
44       - Action Control: xxx
45       - Error Recognition and Correction: xxx
46       - Creative Attempts: xxx
47       - Task Completion Efficiency: xxx
48       - Material Selection and Usage: xxx
49
50   Notes:
51
52   - If the evaluation rules include "e.g.," it is only an example and you
         should not be limited to the listed "e.g." All phenomena that conform
          to the major criteria should be considered.
53
54   - Task progress considers only the completion of key steps of the task
         and is unrelated to artistic qualities or similar aspects.
```

Listing 3: Prompt for individual video rating

## G.7 PSEUDO-CODE EXAMPLES

```javascript
const doc = yaml.load(fs.readFileSync(task_conf, 'utf8'));
// Extract the item name from the task description
const item_name = task_description.split('craft_a_')[1];
// Execute each initialization command to set up the environment
doc.custom_init_commands.forEach(command => {
    bot.chat(command);
});
// Find the recipe for crafting the specified item
const recipe = bot.recipesFor(item_name, craftingTable);
// Attempt to craft the item
try {
    await bot.craft(recipe, count, craftingTablePosition);
    console.log(`${count} ${item_name} crafted successfully`);
} catch(err) {
    console.error('Failed to craft item:', err);
}
```

Listing 4: Mineflayer Craft Task Pseudo-Code

```python
from mcu_benchmark import MinecraftWrapper, VLM_Evaluator
from utility import load_config, check_success_and_save_video
from models import agent_creator

# Step 1: Load task configuration for the benchmark
config = load_config("build_house.yaml")
# Step 2: Initialize the environment with MinecraftWrapper
env = MinecraftWrapper(config['env'], level=config['level'])
# Step 3: Initialize the agent (using custom model path and weights)
agent = agent_creator(model_path, weight_path).cuda()
agent.eval()  # Set the agent to evaluation mode
# Step 4: Get the initial state for the agent
state = agent.initial_state()
# Step 5: Start the environment and reset
obs, info = env.reset()
terminated, truncated = False, False
rollout_info = []
# Step 6: Agent's rollout
while not terminated and not truncated:
    # Get action from the agent and update state
    action, state = agent.get_action(obs, state)
    # Step the environment with the agent's action
    obs, terminated, truncated, info = env.step(action)
    # Save frames (visual feedback from the environment)
    rollout_info.append(info)
# Check if the agent succeeded in the task programmatically
success, video_path = check_success_and_save_video(rollout_info)
# Step 7: Evaluate the agent using a Vision-Language Model (VLM)
vlm_evaluator = VLM_Evaluator()
vlm_score = vlm_evaluator.evaluate(video_path, 'build_criteria.txt')
print(f"Success: {success}. VLM evaluation score: {vlm_score}")
```

Listing 5: MCU Evaluation Process Pseudo-Code

## G.8 CASE STUDY

The following case clarifies the impact of each metric on evaluating generalization performance. Metrics such as task progress and material selection assess basic task alignment, while action control and task efficiency provide insights into optimization strategies. Error correction and creative attempts, in contrast, measure higher-order generalization skills. These are critical for assessing agents in open-ended and complex scenarios, as they reveal resilience to failure and capacity for novel strategies.

While Video B outperformed Video A across most metrics, the weaknesses in creativity and error correction indicate areas where even high-performing agents fall short. Incorporating tailored training modules and broader tasks emphasizing these dimensions will enhance the benchmark's utility for developing and evaluating generalist agents.

```
Task Progress:
- Video A: The agent collects dirt blocks and places them vertically but
    does not reach a reasonable height.
- Video B: The agent collects dirt blocks, places them vertically, and
    reaches a reasonable height.
result: B is better

Action Control:
- Video A: The agent places some blocks horizontally and in unrelated
    locations.
- Video B: The agent places blocks vertically without unnecessary actions
result: B is better

Error Recognition and Correction:
- Video A: The agent does not correct incorrectly placed blocks.
- Video B: The agent does not make any noticeable errors that need
    correction.
result: B is better

Creative Attempts:
- Video A: The agent does not show any creative attempts.
- Video B: The agent does not show any creative attempts.
result: tie

Task Completion Efficiency:
- Video A: The agent takes a longer time with unnecessary actions.
- Video B: The agent completes the task efficiently without unnecessary
    actions.
result: B is better

Material Selection and Usage:
- Video A: The agent uses dirt blocks but places some blocks horizontally
     and in unrelated locations.
- Video B: The agent exclusively uses dirt blocks and places them
    appropriately.
result: B is better

Overall results:
- Task Progress: B is better
- Action Control: B is better
- Error Recognition and Correction: B is better
- Creative Attempts: tie
- Task Completion Efficiency: B is better
- Material Selection and Usage: B is better
```

Listing 6: Video Comparison Evaluation Results

```
1  **Task Progress:**
2  - Evidence: The agent placed two snow blocks vertically and a carved
       pumpkin on top, but no Snow Golem was created.
3  - Score: Partially
4
5  **Action Control:**
6  - Evidence: The agent placed multiple unnecessary snow blocks around the
       structure.
7  - Score: Barely
8
9  **Error Recognition and Correction:**
10 - Evidence: The agent did not correct the placement of the carved pumpkin
       after failing to create a Snow Golem.
11 - Score: Barely
12
13 **Creative Attempts:**
14 - Evidence: No creative attempts or decorations observed.
15 - Score: None
16
17 **Task Completion Efficiency:**
18 - Evidence: The agent took excessive time with unnecessary placements and
       failed to complete the task.
19 - Score: Barely
20
21 **Material Selection and Usage:**
22 - Evidence: Correct materials (snow blocks and carved pumpkin) were used,
       but not effectively.
23 - Score: Partially
24
25 **Overall Scores:**
26 - Task Progress: Partially
27 - Action Control: Barely
28 - Error Recognition and Correction: Barely
29 - Creative Attempts: None
30 - Task Completion Efficiency: Barely
31 - Material Selection and Usage: Partially
```

Listing 7: Individual Video Evaluation Results

