# OpenReview forum: "Towards Evaluating Generalist Agents: An Automated Benchmark in Open World"
_ICLR.cc/2025/Conference — Submitted to ICLR 2025_

### Official Review · Reviewer_WVad · 2024-11-03

**Soundness:** 2
**Presentation:** 2
**Contribution:** 2
**Rating:** 6
**Confidence:** 4

**Summary:**

This paper introduces the MineCraft Universe (MCU), which includes (1) over 3,000 atomic tasks (goals) meant to test basic skills in Minecraft, (2) an LLM-based approach to generating initial states from which the tasks should be executed (aka scene generation), and (3) a VLM-based approach to evaluate whether an agent has successfully completed a task. The goal of this new benchmark is to enable better evaluation of generalist agents that can accomplish a wide variety of goals in Minecraft.

**Strengths:**

1. Minecraft is an excellent domain in which to test the capabilities of generalist agents, and this is an area that should be of increasing importance over the next few years.
2. The VLM evaluator is a significant contribution: earlier benchmarks in Minecraft were often limited to what could be evaluated with programmatic rewards (e.g. MineRL Diamond), or required expensive human evaluation that limited adoption (e.g. MineRL BASALT).
3. The authors have taken steps to increase the quality of their instruction set, beyond that of e.g. MineDojo.
4. There are evaluations of existing Minecraft agents.

**Weaknesses:**

## Evaluation

The significance of the paper depends on claims like:

1. The VLM evaluator provides a robust enough signal that it is useful for evaluating agent performance, and it is better than other comparably cheap evaluation signals in the literature.
2. The suite of atomic tasks / instructions forms a better training or evaluation set for generalist agents than other benchmarks in the literature.
3. It is important that the initial state distribution for an individual atomic task is diverse. (This is needed to justify the utility of the LLM scene generation.)

The paper provides decent evidence for claim (1) in Tables 2 + 3, and maybe Figure 4 too, though it would be nice if it also provided interrater agreement (e.g. taking one human’s labels as ground truth, how well does a different human rater perform), to assess how close the VLM is to human performance.

However, I think the justification for claims (2) and (3) are lacking. For claim (2), the obvious comparison is to MineDojo. Figure 2 presents some flaws with the MineDojo instructions, but it isn’t clear whether this is a major problem – perhaps it is sufficient to use an LLM to automatically filter the MineDojo instructions to remove duplicates and only keep ones at a certain difficulty level. To provide better justification for claim (2), the authors could:

* Cluster the tasks in MineDojo vs MCU, and qualitatively compare the clusters to argue for the benefit of MCU – for example, perhaps when restricting to tasks that are of the appropriate difficulty level, MCU has much larger diversity.
* Build a pipeline to filter MineDojo instructions for appropriate quality, and show that the number of such instructions is much lower than present in MCU. (It would be important to validate the pipeline, though this could be relatively simple, e.g. reporting ten randomly sampled rejected MineDojo instructions in the paper, and arguing that they are indeed problematic.)

The paper doesn’t present any evidence for claim (3). To fix this, the authors could:

* Run evaluations of existing Minecraft agents with (a) a single initial state for each task, and (b) the full initial state distribution for each task. Ideally, we would see qualitatively different conclusions with (b) than with (a), demonstrating that the initial state distribution reveals additional information not present when using a single initial state.
* Finetune a Minecraft agent on a suite of tasks with (a) a single initial state for each task, and (b) the full initial state distribution for each task. Ideally, we would see that (b) produces a much more robust agent.

It would also be good to evaluate the verification pipeline from Mineflayer – for example by reporting what fraction of LLM-generated scenes are rejected by Mineflayer, and using human review to estimate what fraction of those are false positives (i.e. the scenes were possible, but Mineflayer was unable to accomplish the task). However this is less important than the above evaluations.

## Presentation

There are many key details that are necessary to understand and evaluate the paper that are not present, such as:

1. Details on how the 3,000+ atomic tasks were collected, and what quality control was done to ensure these tasks are of high quality. (Particularly important since the authors claim this as a benefit over MineDojo.)
2. An explanation of the Mineflayer verification pipeline. My understanding is that Mineflayer is a programmatic bot, whereas the atomic tasks are written in natural language, so how do you provide the atomic task as an input to Mineflayer to verify that the task is possible?
3. How VPT was evaluated on MCU – to my knowledge, VPT does not take natural language instructions as an input, so it is unclear to me how the task was communicated to VPT. (Perhaps the task was _not_ communicated to VPT, and instead the VPT model was placed in the environment and we simply see if it happens to solve the task in the course of acting – if so, this should be explicitly stated in the main text.)
4. Prompts for the LLM scene generation and VLM evaluation.

Of course, the paper has many components and so there is not enough space to provide full detail about all the components – but the information should still be present in an appendix. I looked through the appendices and didn’t see any of this information, though I might have missed it.

## Significance

Due to the weaknesses in evaluation and presentation mentioned above, it was hard to evaluate how significant the contributions in this paper are.

The authors emphasize that the atomic tasks evaluate basic skills rather than more long-horizon, agentic tasks. It is unclear to me that this is the right focus – it seems possible that with moderately high-level action spaces (as opposed to raw keyboard / mouse inputs), a frontier vision-language model finetuned for Minecraft would saturate this benchmark in the near future. Of course, the benchmark may still be interesting for the development of agents with more low-level action spaces even after that point, as a proxy for real world tasks that require this (e.g. robotics).

Overall I feel very conflicted about this paper. There has clearly been a lot of good work put into it, and some clear contributions such as the VLM evaluator, but I found it difficult enough to assess what was done and how useful it was that I’m recommending a borderline reject.

## Simple fixes

Figure 4: This figure would be much easier to interpret if the x-axis were sorted in order of increasing human scores. (This would allow the reader to easily check that the VLM scores are also increasing on average.)

Line 150: soverability → solvability

Line 267: Add a citation for Mineflayer

Line 373: Specify in more detail which model you use as VPT(bc) – there are several in [Baker et al 2022](https://arxiv.org/pdf/2206.11795). Is it the VPT foundation model finetuned on earlygame_keyword, or on contractor_house, or both, or something else entirely?

Line 507: The Guss et al citation is for the MineRL Diamond competition; the correct citation for MineRL BASALT is [Shah et al 2021](https://arxiv.org/abs/2107.01969).

Line 1182: Malformed figure reference

**Questions:**

1. How does the automatic verification pipeline with Mineflayer work? In particular, given one of your over 3,000 atomic tasks, and an LLM-generated initial state configuration, how do you create a Mineflayer bot that can solve the task (assuming it is possible)?
2. Can you say more, ideally with quantitative evaluations, about why the set of atomic tasks in MCU is better than the tasks in MineDojo?
3. How did you collect the atomic tasks in MCU? How did you ensure their quality?
4. How do you provide task instructions to the VPT(bc) and VPT(rl) methods?
5. Why doesn’t Table 4 report numbers for VPT(rl)?
6. What is the interrater agreement for human evaluations?

---

> ### Author Response · Authors · 2024-11-20
>
> Thank you for your detailed review and recognition of our work. We will now address your questions one by one.
>
> > What is the interrater agreement for human evaluations?
>
> Thank you for the insightful guidance. We fix one human's label as ground truth, and assess how other individuals correspond to this standard across various dimensions. The VLM demonstrates a high level of concordance with human ratings.
> | | Task Progress | Action | Error Recog. | Creative | Efficiency | Material | Average |
> |--------|---------------|--------|--------------|----------|------------|----------|---------|
> | GPT4o | 80.0          | 96.0   | 86.0         | 100.0    | 92.0       | 91.0     | 90.8    |
> | Other-individual | 91.0          | 97.0   | 93.0         | 94.0    | 99.0       | 89.0     | 93.8    |
>
> > Finetune a Minecraft agent on a suite of tasks with (a) a single initial state for each task, and (b) the full initial state distribution for each task. Ideally, we would see that (b) produces a much more robust agent.
>
> Fine-tuning a Minecraft agent remains challenging due to the absence of a standardized pipeline for open-ended policy fine-tuning, and we have not found any relevant literature on this topic. However, the point you raised is analytically well-founded—even without experimental evidence, broadening the state distribution during training is theoretically expected to improve generalization in machine learning.
>
> > How did you collect the atomic tasks in MCU? How did you ensure their quality?
>
> 1. Benchmarking Against Existing Tasks: We filtered tasks from established benchmarks such as Skill-Forge and MineDojo, ensuring that our tasks include representative challenges that have been validated through previous research efforts.
> 2. Minecraft Wiki Resources: Tasks were also collected from the official Minecraft wiki, covering a wide range of activities and interactions present within the game, to capture diverse task scenarios.
> 3. Task Categorization and Item Combinations: The Minecraft simulator we built can return a comprehensive set of environmental information corresponding to common task types, such as crafting, mining, and combat. These tasks are further refined by considering various item combinations.
> 4. LLM and Expert Brainstorming: In addition to benchmark tasks, we conducted brainstorming sessions with domain experts and used large language models (LLMs) to generate novel task ideas that reflect real-world agent challenges.
>
> Since most tasks are from the official Minecraft Wiki and simulator responses, they are inherently reliable. Despite this, all tasks underwent rigorous validation through human inspection and automated scripts.
>
> > Figure 2 presents some flaws with the MineDojo instructions, but it isn’t clear whether this is a major problem – perhaps it is sufficient to use an LLM to automatically filter the MineDojo instructions to remove duplicates and only keep ones at a certain difficulty level. To provide better justification.
>
> Thank you very much for your suggestions. Figure 2 merely serves as an example to demonstrate that our configuration is well-organized and executable. For detailed comparative dimensions, please refer to Table 1, where we have compared our approach with existing benchmarks across multiple dimensions, including environmental, task, and evaluation levels. We have also taken your advice and put both the MCU atomic list and the MineDojo list into GPT, filtering for tasks of a specific difficulty level: "tasks solvable by players with three months of experience, excluding redstone circuits, advanced equipment crafting, and scarce resource acquisition." The final results show 173 tasks from MineDojo and 1,875 tasks from MCU.
>
> > Prompts for the LLM scene generation and VLM evaluation.
>
> In the revised paper, we have included relevant prompt examples in Appendix G.4 and G.5.

---

> ### Author Response · Authors · 2024-11-20
>
> > An explanation of the Mineflayer verification pipeline. My understanding is that Mineflayer is a programmatic bot, whereas the atomic tasks are written in natural language, so how do you provide the atomic task as an input to Mineflayer to verify that the task is possible?
>
> Thank you for your question. Here’s an example of how the automatic verification pipeline operates for a crafting task, using "craft a wooden pickaxe" as the task description and task_conf as the initial state configuration (see Figure 2 for format). The Mineflayer pseudo-code below illustrates this process. We have added it in the appendix of our revised paper.
> ```java
> const doc = yaml.load(fs.readFileSync(task_conf, 'utf8'));
> // Extract the item name from the task description
> const item_name = task_description.split('craft_a_')[1];
> // Execute each initialization command to set up the environment
> doc.custom_init_commands.forEach(command => {
>     bot.chat(command);
> });
> // Find the recipe for crafting the specified item
> const recipe = bot.recipesFor(item_name, craftingTable);
> // Attempt to craft the item
> try {
>     await bot.craft(recipe, count, craftingTablePosition);
>     console.log(`${count} ${item_name} crafted successfully`);
> } catch(err) {
>     console.error('Failed to craft item:', err);
> }
> ```
> For other tasks such as mining and finding, Mineflayer also provides corresponding interfaces like bot.collectBlock.collect and bot.findBlocks to address these issues. Hence, all we need to do is to import tasks and invoke the relevant interfaces to enable batch verification of tasks.
>
> > How do you provide task instructions to the VPT(bc) and VPT(rl) methods?
>
> Thank you for your advice. We will incorporate this part into the revised version. VPT is an unconditioned model, so we didn't provide any instructions and simply observed whether it managed to accomplish the task during its operations.
>
>
> > Why doesn’t Table 4 report numbers for VPT(rl)?
> Sorry, we missed including this part of the experimental data; the new version has been updated with this data.
>
> | Metric | Task Progress | Action | Error Recog. | Creative | Efficiency | Material | Average |
> |--------|---------------|--------|--------------|----------|------------|----------|---------|
> | Vpt-rl | 34.61 | 31.50 | 10.31 | 3.62 | 23.43 | 28.25 | 21.97 |
>
> > Run evaluations of existing Minecraft agents with (a) a single initial state for each task, and (b) the full initial state distribution for each task.
>
> Thank you for your insightful question. In our experiments, we explicitly discuss the difference between using a single initial state versus the full initial state distribution for each task. This is achieved by leveraging our tunable difficulty settings (simple, medium, and hard) to highlight the impact of initial state variability on agent performance.
>
> Our results demonstrate that difficulty levels inherently capture the diversity in initial states. For example, tasks categorized as "hard" include more complex initial conditions, such as increased environmental distractions or additional items, which significantly challenge the agents compared to "simple" setups. As shown in our evaluation (Figure 6, Section 4.2.3), agent performance declines sharply as task complexity increases, underscoring how richer initial state distributions reveal limitations in generalization and robustness that are not apparent with a single state. These findings validate that the full initial state distribution provides critical insights into agent capabilities, emphasizing its importance for comprehensive benchmarking.
>
>
> > Significance: The authors emphasize that the atomic tasks evaluate basic skills rather than more long-horizon, agentic tasks. It is unclear to me that this is the right focus. Overall I feel very conflicted about this paper. There has clearly been a lot of good work put into it, and some clear contributions such as the VLM evaluator, but I found it difficult enough to assess what was done.
>
> Thank you for your positive feedback on our VLM work. I understand your concerns and I'm glad to see that our responses have addressed some of them. To clarify, our work is not solely focused on atomic tasks. Instead, MCU can handle complex, multi-step tasks through task initialization and composition, allowing the combination of atomic tasks into long-term tasks that require advanced planning. 1. Task initialization: MCU allows different task initializations, where tasks like mining diamonds can be approached at varying levels of complexity by controlling the initial environment—e.g., providing a tool vs starting from scratch. 2. Composition Rules (Lines 204-209): We propose composition rules to guide the creation of multi-step tasks. Despite current controller limitations causing even advanced agents to struggle with atomic tasks, our method is readily adaptable for more complex, long-horizon tasks.

---

> > ### Comment · Reviewer_WVad · 2024-11-24
> > **Raising score**
> >
> > Thanks for the additional details and the answers to the questions. I've read the rebuttal and the other reviews and am raising my score to 6 -- please do make sure to include the additional details in the paper.

---

> ### Author Response · Authors · 2024-11-26
>
> We are delighted that our rebuttal has addressed your concerns and truly appreciate your recognition of our work. We will continue to strive and make progress on this path.

---

### Official Review · Reviewer_KdK1 · 2024-11-04

**Soundness:** 2
**Presentation:** 2
**Contribution:** 2
**Rating:** 3
**Confidence:** 1

**Summary:**

This paper presents a new agent evaluation benchmark based on the minecraft environment.

**Strengths:**

- **New incremental improvement over existing benchmarks**: As shown in Table 1 and Figure 2, the proposed benchmark brings clear improvement over [Minedojo](https://github.com/MineDojo/MineDojo), by introducing open-ended tasks, simulation on real game playing, and various difficulty level. The pipeline with automatic task generation and automatic evaluation is also a nice contribution. I believe the engineering efforts could be non-trivial, if reproducible.

**Weaknesses:**

- **Overclaims**: I am concerned about potential overclaims made by the authors. I suggest the authors to adjust the tone for more truthful/precise/rigorous presentation. For example:
    - The evaluation environment relies solely on Minecraft. I suggest the authors to explicitly add Minecraft in the title or reflect the ''Open World'' is a simulated one. Additionally, is `generalist agents` well defined and well captured by this specific environment in the paper? I lean towards **no**. As I see it, a more precise description might be `open-ended embodied agents`. I recommend the authors to take a look at [Hughes et al., 2024](https://arxiv.org/abs/2406.04268), where open-endedness is precisely defined. If the authors really want to use ''generalist'', it would be more convincing to include multiple different or realistic environments beyond a single game, like [Gato](https://deepmind.google/discover/blog/a-generalist-agent/) / [AgentBench](https://github.com/THUDM/AgentBench) / [VisualAgentBench](https://github.com/THUDM/VisualAgentBench) / [CRAB](https://github.com/camel-ai/crab) / [SWE-Bench-M](https://www.swebench.com/multimodal.html).
- **Experimentation**:
    - Are all the agents evaluated pre-trained open-source models? Can one use API-based agents (e.g., gemini, gpt) with MCU?
    - How easy can people use the proposed benchmark? Could the author provide code to illustrate? How long does it take for a full evaluation and how much resources would cost?
    - Would you provide a more detailed and clear description for the human rating system in both the main text and appendix?
    - How robust is the evaluation? For example, can the authors try different LLM judges and see if there are any qualitative differences?
- **Writing**: Please improve the writing clarity. I find the paper sometimes hard to follow. (Minor note: many citations (e.g., line 49), section ref (e.g., line 148) and marks (you should use `' instead of "" in latex) are not in the correct format.) Please also consider adding a section on limitations.


References:
- Hughes, Edward, et al. "Open-Endedness is Essential for Artificial Superhuman Intelligence." arXiv preprint arXiv:2406.04268 (2024).

**Questions:**

Please see the weaknesses part.

I am concerning about the reproducibility of this work and the veracity of the claims. I am happy to raise my score if the authors can make concrete efforts to address the concerns.

**Details Of Ethics Concerns:**

I am concerning about the potential research integrity of this submission (e.g., reproducibility). While the authors propose an automated benchmarking framework, the pseudo-code provided in G.7 does not adequately support reproducibility and appears too abstract to serve as a reliable guide for implementation. As it stands, the level of detail is insufficient to validate the claims made in the paper or enable others in the community to replicate the proposed methods effectively. If the authors do not intend to open-source the code, I suggest providing a more detailed and testable version of the pseudo-code to substantiate their claims and ensure that others can effectively reproduce the results. Without such efforts, accepting the paper may risk undermining the standards of reproducibility and accountability in this research community.

---

> ### Author Response · Authors · 2024-11-20
>
> Thank you for your valuable feedback and insightful questions.
>
> > Overclaims: I am concerned about potential overclaims made by the authors. I suggest the authors to adjust the tone for more truthful/precise/rigorous presentation. For example: The evaluation environment relies solely on Minecraft. I suggest the authors to explicitly add Minecraft in the title or reflect the ''Open World'' is a simulated one. Additionally, is generalist agents well defined and well captured by this specific environment in the paper? I lean towards no. As I see it, a more precise description might be open-ended embodied agents. I recommend the authors to take a look at Hughes et al., 2024, where open-endedness is precisely defined. If the authors really want to use ''generalist'', it would be more convincing to include multiple different or realistic environments beyond a single game, like Gato / AgentBench / VisualAgentBench / CRAB / SWE-Bench-M.
>
> Thank you for your thoughtful suggestion. We are happy to adjust the tone to better align with the scope of our work. Regarding the concept of a generalist agent, we acknowledge that our definition has been more descriptive rather than the rigorous framework provided by Hughes et al., in the context of open-endedness. As this work represents only a small step toward the development of generalist agents, we are committed to clarifying any claims that might obscure our contributions (e.g., simulated open worlds, open-ended embodied agents) in the revised paper.
>
> > Are all the agents evaluated pre-trained open-source models? Can one use API-based agents (e.g., gemini, gpt) with MCU?
>
> Yes, all the agents we have evaluated are based on pre-trained, open-source models. Additionally, our system supports API-based agents, such as Gemini or GPT. These agents are compatible with our system as long as they align with the observation space, which consists of screen-pixelated images, and the action space as defined in the human play settings (refer to Table 6)." Our system is model-agnostic and policy-agnostic, ensuring compatibility with a wide range of agent types.
>
> > How easy can people use the proposed benchmark? Could the author provide pseudo code to illustrate? How long does it take for a full evaluation and how much resources would cost?
>
> **Ease of Use**:
> Our proposed benchmark is designed to be user-friendly, allowing individuals to easily interact with it through our provided interface. We have included pseudo-code in the revised paper (Listing 5: MCU Evaluation Process Pseudo-Code) to further illustrate the process. This pseudo-code serves as a clear guide for users to understand how to utilize the benchmark effectively.
>
> **Flexibility and Customization**:
> In addition to the ease of use, we have developed a flexible interface that supports the training and testing of text-conditioned, video-conditioned, and unconditioned agents. This interface also includes personalized features such as a reward function, enabling users to design customized reward functions for open-ended tasks in the future.
>
> **Resource Consumption**:
> Regarding resource consumption, using the GROOT benchmark as an example, running inference across 35 tasks on a single A40 GPU takes approximately 30 minutes. The VLM evaluation process is completed in about 10 minutes.
>
> > Would you provide a more detailed and clear description for the human rating system in both the main text and appendix?
>
> Certainly! We've added a detailed description of our human rating system in Appendix G.3, including screenshots of the rating interface. Each component's function is briefly explained to provide clarity. Due to time constraints, we can't cover every detail, but we're happy to expand on it in future communications.
>
> >  How robust is the evaluation? For example, can the authors try different LLM judges and see if there are any qualitative differences?
>
> We have expanded Table 2 to include different models, such as the open-source model minicpm and OpenAI's gpt4-o-mini. The numbers represent F1 scores for classifying the better trajectory. While there is a gap between the open-source model minicpm and commercial models, the results from gpt4-o-mini suggest that the performance gap is narrowing. As the capabilities of vision-language models (VLMs) enhance, our method is likely to show even greater potential.
>
> | Model | Survive | Build | Craft | Tool | Collect | Explore | Average |
> |-------|---------|-------|-------|------|---------|---------|---------|
> | MineClip | 11.0 | 45.0 | 44.0 | 44.0 | 73.0 | 0.0 | 44.0 |
> | Ours(minicpm) | 65.0 | 33.0 | 80.0 | 75.0 | 33.0 | 53.0 | 66.0 |
> | Ours(gpt4o-mini) | 73.0 | 58.0 | 86.0 | 80.0 | 40.0 | 75.0 | 75.0 |
> | Ours(gpt4o) | 100.0 | 85.0 | 62.0 | 58.0 | 73.0 | 100.0 | 80.0 |

---

> ### Author Response · Authors · 2024-11-20
>
> > If the authors do not intend to open-source the code, I suggest providing a more detailed and testable version of the pseudo-code to substantiate their claims and ensure that others can effectively reproduce the results.
>
> To address your concerns regarding reproducibility, we have uploaded the code for our MCU benchmark in the supplementary materials. In the interest of keeping the codebase concise, we have removed certain simulator-related code. However, the detailed steps for inference and automatic evaluation are clearly outlined in the README file. We encourage you to refer to it for further clarification.

---

> > ### Comment · Reviewer_KdK1 · 2024-11-24
> >
> > Thank you very much for your time and the well-written rebuttal. I have taken another pass on the submission. Please see below for my additional concerns:
> >
> > 1. Minor: The paper may still contain some overclaims. Besides the generalist agents perspective mentioned earlier, there are other overclaims, such as "maximal freedom" etc. I suggest the authors avoid using these words. Academic writing is not about crafting captivating fiction, but more about presenting precise, rigorous and well-founded research.
> >
> > 2. I am unable to reproduce the results reported by the authors with the provided code at this stage, thus I cannot verify the veracity of the reported results. Also, I suggest the authors carefully check the code before submitting for review to ensure it is both correct and fully anonymized (e.g., no dir paths that may potentially reveal authors' identity/affiliations).
> >
> > 3. The description of the human subject studies is not convincing enough to me. The potential lack of transparency reduces confidence in the evaluation process. I have lowered my confidence score accordingly.
> >
> > 4. The manuscript has many presentation issues and may not be ready for publication: overclaims (see point 1), missing refs, typos, and insufficient literature review (I would also suggest the authors include a thorough literature review in the appendix to better contextualize their work).
> >
> > 5. Regarding the literature review, I am surprised by the authors' review on LLM-as-Judge. Quote below:
> >
> >    > "**LLM-as-Judge**. The feasibility of using Large Language Models (LLMs) as judges lies in their ability to process and analyze vast amounts of **legal data**, making them suitable for tasks requiring nuanced understanding and reasoning Manakul et al. (2023); Touvron et al. (2023). As highlighted in recent studies, LLMs like GPT-4 achieve high agreement rates with human evaluators in **legal reasoning tasks** Zheng et al. (2023); Liu et al. (2023); Li et al. (2023), demonstrating their potential in replicating human judgment in complex cases. These models provide scalability, cost-efficiency, and consistency, addressing challenges in the **legal domain** **[Stanley et al. (2017)](https://www.oreilly.com/radar/open-endedness-the-last-grand-challenge-youve-never-heard-of/); [Standish (2003)](https://arxiv.org/pdf/nlin/0210027)**, such as bias and decision fatigue, making them valuable in **judicial systems**."
> >
> >    The entire paragraph is about legal stuffs, which has little-to-no relevant to the real/practical meaning of llm-as-a-judge. The original llm-as-a-judge paper (Zheng et al., 2023) did not even mention a single word of legal—it is about evaluating instruction-following and on alignment with humans. Citing irrelevant works like Stanley et al. (2017) and Standish (2003) is also careless (the two works are about open-endedness and have zero relevance to legal domains and llm-as-a-judge). The paragraph is thus irrelevant and misleading (the first sentence "... lies in their ability to ... analyze legal data" is not true). The authors should be more responsible for the accuracy and relevance of their claims when presenting their work to fellow researchers.
> >
> > 6. Overall, this submission gives the impression of an unfinished piece of work wrapped in a fancy package. I strongly encourage the authors to prioritize delivering **usable, trustworthy and reproducible** content in future revisions—something that can genuinely benefit the research community and advance the field.

---

> ### Author Response · Authors · 2024-11-26
>
> Thank you for your response. We will address each of your remaining concerns one by one.
>
> > Minor: The paper may still contain some overclaims. Besides the generalist agents perspective mentioned earlier, there are other overclaims, such as "maximal freedom" etc.
>
> Thank you for your reminder. We have removed subjective phrases like “maximal freedom” and refined our claims for greater precision and rigor in the revised paper.
>
> > I am unable to reproduce the results reported by the authors with the provided code at this stage, thus I cannot verify the veracity of the reported results. Also, I suggest the authors carefully check the code before submitting for review to ensure it is both correct and fully anonymized (e.g., no dir paths that may potentially reveal authors' identity/affiliations).
>
> Thank you for your suggestions. We have ensured that all the code is anonymized, and we appreciate your reminder on this matter. Additionally, we have provided a more refined version of the code, which includes more executable task initialization configurations. We believe you should be able to run automatic generation and evaluation following the README file (please remember to change the OpenAI key).
>
> > The description of the human subject studies is not convincing enough to me. The potential lack of transparency reduces confidence in the evaluation process. I have lowered my confidence score accordingly.
>
> We understand your concerns, and here is a detailed introduction to human subject studies:
>
> We collected 562 data entries: 236 from comparative video ratings and 226 from single video ratings. Tables 2 and 3 are derived from the comparative ratings, where humans evaluated videos for the same task based on performance dimensions. Confusing cases could be skipped. Table 2 shows F1 scores by task type, demonstrating notable improvement over the MineDojo model. Table 3 reports alignment scores across rating dimensions. The results in Figure 4 are based on 226 samples randomly selected from our video dataset. Tasks with fewer than 3 human annotations were excluded to ensure data reliability. For the remaining tasks, we calculated the mean of human ratings and VLM assessments to present the results. All the VLM results experienced a minor reduction in information due to frame sampling (every 30 frames).
>
> We acknowledge that we did not collect tens of thousands of data points or conduct an exhaustive evaluation of the VLM evaluator’s capabilities, as is typical in papers dedicated to LLM alignment [1, 2]. However, our work does not primarily aim to advance auto-evaluation methods alone. Instead, we propose an automated benchmarking pipeline for open-ended tasks as a whole. Within this broader context, our auto-evaluation method, supported by single-rating and comparison-rating system, demonstrates reliability and effectiveness, as evidenced by improvements over the MineClip baseline. Furthermore, its performance is expected to improve alongside advancements in VLMs, reinforcing the utility of our pipeline for future research.
>
>
> > The manuscript has many presentation issues and may not be ready for publication: overclaims (see point 1), missing refs, typos, and insufficient literature review (I would also suggest the authors include a thorough literature review in the appendix to better contextualize their work).
>
> We have thoroughly revised the paper to address typos, incomplete references, and missing citations. The literature review will be significantly expanded in the appendix to contextualize our work comprehensively.
>
>
> > Regarding the literature review, I am surprised by the authors' review on LLM-as-Judge.
>
> Sorry for the inaccuracies in our related work section. We've revised it and would like to get your feedback on the updated paper.
>
>
> References:
>
> [1] Liu Y., Iter D., Xu Y., et al. G-Eval: NLG Evaluation Using GPT-4 with Better Human Alignment. arXiv preprint arXiv:2303.16634, 2023.
>
> [2] Wang J, Liang Y, Meng F, et al. Is chatgpt a good nlg evaluator? a preliminary study[J]. arXiv preprint arXiv:2303.04048, 2023.

---

> ### Author Response · Authors · 2024-11-26
>
> > Overall, this submission gives the impression of an unfinished piece of work wrapped in a fancy package. I strongly encourage the authors to prioritize delivering usable, trustworthy and reproducible content in future revisions—something that can genuinely benefit the research community and advance the field.
>
> Thank you for your valuable suggestions. While we acknowledge your concerns regarding the maturity of our work, we respectfully disagree with the characterization of our submission as "unfinished." Below, we outline the key contributions and the necessity of our work, which we believe directly address the challenges faced by the community and provide meaningful advancements:
>
> **Key Contributions:**
> 1. Composable atomic tasks:
>   We have collected over 3,000 atomic, composable tasks, offering a diverse set of challenges. These include tasks such as Trade (logical reasoning), Mining (physical interaction), Combat (strategic planning), Building (artistic creation), and Trapping (precision control), which allows researchers to explore a wide range of testing scenarios.
> 2. Task generalization and verification pipeline:
>   By leveraging LLMs, each task is dynamically generated and uniquely instantiated during each evaluation to promote essential generalization skills in agents. A validation pipeline ensures tasks are logically correct and solvable, providing executable initialization configurations to the community.
> 3. Domain-general evaluator:
>   We provide a domain-general Vision-Language Model (VLM) evaluator, capable of dynamically assessing open-ended tasks across six key dimensions (e.g., task progress, creativity). This introduces a novel, flexible evaluation mechanism tailored to open-world benchmarks.
> 4. Automatic benchmarking:
>   MCU leverages LLM-driven task generation and a VLM-based evaluator to establish a scalable and automated benchmarking system, which eliminates reliance on human annotations while improving reproducibility and accessibility for large-scale assessments.
>
>
> **Necessity of MCU:**
>
> 1. Addressing benchmark gaps: MCU fills these gaps with an automatic benchmarking pipeline, facilitating research in complex, open-world environments.
>
> 2. Promoting generalization: Experiments reveal that current state-of-the-art agents struggle with task complexity and adaptability, underscoring MCU’s role in fostering robust agent development.
>
> 3. Scalability and efficiency: The automated processes enable large-scale, multidimensional evaluations with minimal human intervention, offering a cost-effective and scalable alternative to traditional methods.
>
>
> While we recognize that certain aspects of MCU, such as enhancing the exploration of longer-horizon tasks, could be further enriched in future work, we believe that this submission lays a solid foundation for automatic benchmarking in open-world environments. It provides valuable tools and insights that genuinely advance the field and address critical research gaps. Also, we will open-source our code and continually improve the benchmark to ensure usability and customization, fostering greater community collaboration and impact.

---

> > ### Comment · Reviewer_KdK1 · 2024-11-26
> > **On Respectful Disagreement**
> >
> > # Thank you for your reply. Regarding the last point, by the time that I submitted my previous review, \**the submitted code is incomplete and not runnable\** / there were \**no trustworthy reproducibility statements\** / the literature review contained \**false and misleading claims\** (more than *inaccuracies* as downplayed in the rebuttal) / the human evaluation section was vague. All those are serious concerns (Please check the authors' guide on reproducibility on "It is important that the work published in ICLR is reproducible.").
> >
> > This has led me to perceive the work as “an unfinished piece of work wrapped in a fancy package.” Unfortunately, the rebuttal appears to strategically downplay or gloss over these critical issues (e.g., reproducibility and the veracity of claims).
> >
> > Academic presentation is not about selling a stuff, but rather educating the audiences with truths and enriching the knowledge of the community. There could be imperfection in a submission, but as reviewers and authors we should collaborate to improve it better *while upholding integrity*. I would review the updated manuscript and code to see if they may guarantee an improved score. In the meantime, please feel free to provide any reproducibility instructions that you believe to be helpful.

---

> > > ### Comment · Reviewer_KdK1 · 2024-12-03
> > >
> > > I have revised my score given the final manuscript. I cannot give a higher score due to the above unresolved concerns on veracity of claims, reproducibility, and paper writing — I would encourage the authors to make concrete efforts enhancing these aspects in future revisions. Adding support for other commonly used multimodal llms like gemini/gpt for the agents in the implementation should also make the package more useful and impactful, as mentioned in the beginning.

---

> ### Author Response · Authors · 2024-12-03
>
> Thank you for your reply. We think we have revised all the unclear claims in the paper. Regarding reproducibility, we have uploaded runnable code with a README for guidance. This code includes task generation, criteria generation, and the automated evaluation program. Additionally, we have provided a list of over 3,000 atomic tasks, a series of executable initialization configurations, and the corresponding criteria files. The appendix of our paper also includes prompts for both LLMs and VLMs. Our interfaces also support LLM-based agents, as if they were using a mouse and keyboard for input. In the end, we sincerely appreciate all the suggestions you have provided, they will definitely enhance our work.

---

### Official Review · Reviewer_AoEZ · 2024-11-07

**Soundness:** 3
**Presentation:** 3
**Contribution:** 3
**Rating:** 6
**Confidence:** 4

**Summary:**

This paper presents the MineCraft Universe (MCU), an automated benchmarking framework designed to evaluate generalist agents within the open-world environment of Minecraft. The MCU framework aims to push boundaries in evaluating agent generalization capabilities by generating over 3000 atomic tasks that vary across difficulty levels and integrate compositional complexity. Key features include the dynamic generation of tasks using large language models (LLMs), verification mechanisms for solvability, and a vision-language model-based (VLM) evaluation system that assesses multi-dimensional performance metrics. Experiments reveal that existing generalist agents, while proficient in certain tasks, struggle with more complex and creative tasks. The paper argues for MCU’s contribution to developing robust generalist agents that can generalize and interact effectively in open-world scenarios.

**Strengths:**

- Comprehensive Benchmarking Framework: MCU introduces a robust benchmarking approach for testing generalist agents. Its combination of task generation, verification, and evaluation creates a solid foundation for assessing performance in open-ended environments.
- Task Diversity and Complexity: The framework’s extensive task library and compositional complexity promote intra- and inter-task generalization, enabling a nuanced understanding of agent adaptability and scalability.
- Evaluation Consistency with Human Judgment: The paper demonstrates that the automated VLM-based evaluations align closely with human assessments, enhancing MCU’s credibility as a scalable, automated benchmark.
- Real-World Relevance: By leveraging Minecraft's open-ended nature and unpredictability, MCU provides a realistic environment that tests generalist agents' adaptability and robustness in dynamic, real-world scenarios.

**Weaknesses:**

- Missing important references on data generations for agent benchmarking. The atomic task composition approaches for task generation have been explored in previous works like CRAB: Cross-environment Agent Benchmark for Multimodal Language Model Agents
- Limited Validation with State-of-the-Art Agents: While MCU's evaluation was conducted on several agents, the majority of tested models (e.g., VPT, STEVE-1) represent early-stage developments in generalist AI. The framework’s effectiveness in evaluating more recent, advanced models and agents is uncertain.
- Reliance on Minecraft-Specific Dynamics: Although Minecraft provides a useful testbed, the heavy reliance on its specific mechanics may limit generalizability. Results might be biased toward models fine-tuned for Minecraft’s environmental patterns, rather than broader real-world adaptability.
- Insufficient Analysis of Evaluation Metrics: While multi-dimensional evaluation is emphasized, more in-depth analysis of how each metric (e.g., error recognition, creative attempts) impacts generalization performance across task types would clarify MCU's potential in training versus evaluating.

**Questions:**

- What criteria were used for selecting the 3000 atomic tasks, and how representative are they of a wide range of real-world agent challenges?
- Could the MCU framework be adapted to test agents in other open-world environments, or is it largely Minecraft-specific?
- How well does MCU handle highly complex, multi-step tasks that may require advanced planning or problem-solving strategies?

Minor
- Line 49 Missing citation
- Table 1: Missing space between Benchmark names and citations

---

> ### Author Response · Authors · 2024-11-19
>
> Thank you for the insightful review! Our work introduces a robust benchmarking framework for assessing generalist agents, a diverse and complex task library for promoting adaptability, evaluations that align closely with human judgment, and a real-world relevant environment for testing agent robustness. Now, I will answer the questions and describe how we will revise the paper.
>
> > Missing important references on data generations for agent benchmarking. The atomic task composition approaches for task generation have been explored in previous works like CRAB: Cross-environment Agent Benchmark for Multimodal Language Model Agents.
>
> Thank you for the reminder. In our revised paper, we have included a discussion for this article.
>
> >- Limited Validation with State-of-the-Art Agents: While MCU's evaluation was conducted on several agents, the majority of tested models (e.g., VPT, STEVE-1) represent early-stage developments in generalist AI. The framework’s effectiveness in evaluating more recent, advanced models and agents is uncertain.
>
> Thank you for your comment. In fact, we choose to evaluate text-conditioned agents (e.g., STEVE-1), video-conditioned agents (e.g., GROOT), and unconditioned agents (e.g., VPT). These models are among the latest open-source offerings in the field and effectively cover the three primary categories of agent conditioning.
>
> From our perspective, agents like those developed by MineDojo are not publicly accessible, and others, such as DEPS and Voyager, are designed to train controllers that do not depend on general mouse/keyboard inputs (please refer to Appendix G.2 for details). These controllers are typically not available in other environments, which pose challenges for transferring them to diverse settings. We would be happy to incorporate more agents that you suggest would be relevant for our evaluation.
>
> > Reliance on Minecraft-Specific Dynamics: Although Minecraft provides a useful testbed, the heavy reliance on its specific mechanics may limit generalizability. Results might be biased toward models fine-tuned for Minecraft’s environmental patterns, rather than broader real-world adaptability.
>
> We appreciate the reviewer’s insightful comment regarding the potential generalizability of MCU. While it is true that Minecraft offers a unique environment, we believe that the MCU framework is designed with substantial flexibility and generalization potential across open-world environments. Below, we highlight the key reasons:
> 1. LLM-driven Task Generation: The task generation process within MCU leverages LLMs, which can create a wide array of environments and tasks based on broad knowledge. This allows MCU to adapt to various open-world settings beyond Minecraft, as LLMs can handle diverse task generation patterns and incorporate new environmental rules and challenges, ensuring that the framework is not bound by Minecraft-specific constraints.
> 2. Domain-general evaluator: MCU utilizes VLMs for automated task evaluation. This evaluation approach relies on evidence extracted from video analysis, making it platform-agnostic. It does not depend on Minecraft’s mechanics but rather on general principles of task completion and agent performance. This allows MCU’s evaluation framework to be easily applied to other environments with minimal modification.
> 3. Modular and Scalable Design: MCU is architected with modularity in mind. Its components—task generation, validation, and evaluation—are designed to be independent of the specific environment. This modularity facilitates straightforward adaptation to new environments, as only the environment-specific modules (e.g., task generation rules) need to be redefined, without impacting the core logic.
> 4. Open-World Generalization: Since MCU is designed to handle the complexities and uncertainties typical of open-world environments, its core features—such as dynamic task progression, error recognition, and creativity evaluation—are relevant and transferrable to other open-world settings.
> In summary, while Minecraft serves as a convenient testbed, MCU's flexibility, modularity, and task generalization capabilities ensure that the framework can be easily adapted to other open-world environments. We acknowledge that adapting to specific environments may require some effort in task reconfiguration, but the underlying architecture supports seamless migration to diverse settings with minimal additional cost.
>
> > - Insufficient Analysis of Evaluation Metrics: While multi-dimensional evaluation is emphasized, more in-depth analysis of how each metric (e.g., error recognition, creative attempts) impacts generalization performance across task types would clarify MCU's potential in training versus evaluating.
>
> We have outlined a case in Appendix G.8 of the revised paper. Due to time constraints, we have conducted only a basic analysis at this stage. We intend to delve into more in-depth analysis in the subsequent phases.

---

> ### Author Response · Authors · 2024-11-19
>
> > What criteria were used for selecting the 3000 atomic tasks, and how representative are they of a wide range of real-world agent challenges?
>
> We mainly collect tasks from the following four approaches:
> 1. Benchmarking Against Existing Tasks: We filtered tasks from established benchmarks such as Skill-Forge and MineDojo, ensuring that our tasks include representative challenges that have been validated through previous research efforts.
> 2. Minecraft Wiki Resources: Tasks were also collected from the Minecraft wiki, covering a wide range of activities and interactions present within the game, to capture diverse task scenarios.
> 3. Task Categorization and Item Combinations: The Minecraft simulator we built can return a comprehensive set of environmental information corresponding to common task types, such as crafting, mining, and combat. These tasks are further refined by considering various item combinations.
> 4. LLM and Expert Brainstorming: In addition to benchmark tasks, we conducted brainstorming sessions with domain experts and used large language models (LLMs) to generate novel task ideas that reflect real-world agent challenges.
>
> To ensure task validity, all tasks underwent rigorous validation through both human inspection and automated scripts.
> The selected atomic tasks span a variety of domains, each reflecting key real-world agent capabilities:
> as we demonstrated in Introduction (line 79-81): Trade (logical reasoning), Mining (physical interaction), Combat(strategic planning), Building (artistic creation), Trapping (precision control), and Redstone (complex-knowledge application), navigation (path planning)
>
> > Could the MCU framework be adapted to test agents in other open-world environments, or is it largely Minecraft-specific?
>
> Please refer to our response above regarding the potential generalizability of MCU.
>
> > How well does MCU handle highly complex, multi-step tasks that may require advanced planning or problem-solving strategies?
>
> MCU handles complex, multi-step tasks through task initialization and composition, allowing the combination of atomic tasks into long-term tasks that require advanced planning.
> 1. Task initialization: MCU allows different task initializations, where tasks like mining diamonds can be approached at varying levels of complexity by controlling the initial environment—e.g., providing a tool vs starting from scratch
> 2. Composition Rules (Lines 204-209): We propose composition rules to guide the creation of multi-step tasks, and show some Complex Task Examples (Lines 972-1002)
> 3. Experimental Validation: We have also conducted experiments with some compositional tasks. Figure 5 illustrates representative compositional tasks, including long-horizon tasks like crafting a table from scratch or combining dyeing and shearing tasks to test the agent's ability to handle complex task sequences.
> As agent capabilities evolve, MCU can scale to handle even more sophisticated long-term tasks, e.g. Fight the Ender Dragon.
>
> > Line 49 Missing citation. Table 1: Missing space between Benchmark names and citations
>
> Thanks for your reminder, We will address these issues in the our paper.

---

### Official Review · Reviewer_CMiP · 2024-11-07

**Soundness:** 2
**Presentation:** 1
**Contribution:** 1
**Rating:** 1
**Confidence:** 5

**Summary:**

The paper introduces MineCraft Universe (MCU), an automated benchmarking framework for evaluating so-called generalist AI agents within Minecraft. MCU consists of three components: a dynamic task generation mechanism, a set of over 3,000 composable atomic tasks, and an evaluation framework supporting open-ended task assessment. The framework evaluates agents across six key dimensions: task progress, action control, material usage, task efficiency, error recognition, and creative attempts. The automated evaluation pipeline can assess open-ended tasks without explicit end states and shows reasonable concordance with human assessments across different dimensions. The authors evaluated VPT, STEVE-1, and GROOT using their framework.

**Strengths:**

1. **Originality.** The paper includes a creative use of LLMs and VLMs for both task generation and evaluation.

2. **Quality.** The empirical validation showing higher agreement between automated and human evaluation is good.

3. **Clarity.** I could get the general gist of what the paper is doing.

4. **Significance.** Minecraft is a very interesting and popular domain of study.

**Weaknesses:**

1. **Lack of code.** I find it strange that it is a D&B paper that does not include anonymized code. This makes it very difficult to answer questions that are not addressed in the paper.

2. **Generally unclear writing.** I found it incredibly difficult to understand the details of the paper. For example, "We hire 20 people with expertise in the field of Minecraft, and annotate 1 hour of data.'' Does this mean that each person annotates one hour of data? As another example, ``These trajectories exhibit highly similar poses for many steps, thereby increasing the evaluative complexity.'' Unclear why this is true?

3. **Incorrect or unclear claims about prior work.** The claim that BEDD does not include a comprehensive evaluation of capabilities is not an accurate characterization of the work (line 506). In fact, one of the key contributions of BEDD is its decomposed evaluation framework, which enables comparison along decomposed subgoals and other characteristics, such as human likeness. Furthermore, the claim that tasks require long-horizon planning, which reduces composability is unclear. Typically, we think about long-horizon tasks in Minecraft as being a composition of other object- or goal-conditioned tasks. As a result, this claim is not well-supported. Furthermore, the authors claim that some prior work fine tunes foundation models for Minecraft, but they do not comment on why these were not studied or are insufficiently general (lines 520-521).

4. **Results presentation lacking.** The figures do not contain error bars. It's also unclear what is being presented in Figure 4. Is this an aggregate of all evaluations? Specific cherry-picked examples? The gaps seem quite large for some evaluations, so I'm a little confused by the Kendall's tau value that is reported.

5. **Unclear benefit of use of LLMs.** The authors claim that a benefit of their approach is that, given the 6 high-level criteria, the LLMs can "autonomously generate tailored criteria for each task''. But it is unclear why this is desirable, and it is also unclear whether the criteria would be the same with each trial. Is there a fixed set of factors for each of the high-level criteria?

6. **Contradictory evaluation?** I don't understand how the agents are even evaluated. In one part of the paper, it says, ``options ranging from ”a is better,” ”b is better,” ”tie,” to ”both are bad.”", yet in another part of the paper, it says, "We define the scoring intervals for each criterion as follows: very poor, poor, fair, good, and excellent." Which one is it? Is it both?

7. **Overclaims without evidence.** The authors claim that the evaluation offers outstanding performance and profound insights (lines 359-361). But I don't see any support for this in the paper.

**Questions:**

1. Line 330: where do the human gameplay videos come from? This is never specified.

2. 4.2.2 essentially shows the same finding as BASALT: compositional tasks that require creativity and/or deep knowledge of Minecraft are still difficult. Could the others clarify how this finding is different?

3. Could the authors clarify why the task definition is tool independent? In Minecraft, it would be meaningful to mine dirt with a diamond shovel over a wooden pickaxe. The former would be more efficient and would require traversing more of the tech tree. So I don't think that this task definition makes too much sense.

4. What are the difficulty settings that get introduced? I don't understand how this is systematically implemented.

**Details Of Ethics Concerns:**

**Use of LLM to write the paper.** It seems like the authors took the VPT, MineDojo, & BEDD papers and put them into an LLM and directly used the outputs. I also believe that lines 523-530 provide concrete evidence that an LLM was used to write the paper. Specifically, this section is supposed to be about LLMs as judges of Minecraft tasks. However, the LLM derails and starts talking about LLMs as judges in **legal cases**. This derailment makes no sense.

**Possible plagiarism.** I have seen a very similar paper on arXiv that introduces the Minecraft Universe (MCU) with striking similarities. Link here: https://arxiv.org/abs/2310.08367. In addition to the name, the similarities are: 3000(+) tasks, 6 evaluation criteria, and the general "vibe" of the paper. But the paper looks like it was put through an LLM before submitting it.

---

> ### Author Response · Authors · 2024-11-19
>
> Thank you for your response and valuable suggestions. We will address each of your concerns one by one.
> > Lack of code. I find it strange that it is a D&B paper that does not include anonymized code. This makes it very difficult to answer questions that are not addressed in the paper.
>
> To address your concerns regarding reproducibility, we have uploaded the code for our MCU benchmark in the supplementary materials. For the sake of clarity, we have removed some simulator-related code. You can refer to the README document for better understanding of the code structure.
>
> >Generally unclear writing. I found it incredibly difficult to understand the details of the paper. For example, "We hire 20 people with expertise in the field of Minecraft, and annotate 1 hour of data.'' Does this mean that each person annotates one hour of data? As another example, ``These trajectories exhibit highly similar poses for many steps, thereby increasing the evaluative complexity.'' Unclear why this is true?
>
> Yes, we hire 20 experts in the field of Minecraft to annotate data, with each person contributing one hour of annotation work.
> About the issue of trajectory similarity, we think we have explained this issue in Lines 330-332. "Unlike the majority of previous work, which typically contrasted successful and unsuccessful trajectories, our dataset predominantly consists of trajectories from similar agents across different rollouts." For example, our evaluator differentiates between trajectories achieving 30% vs. 60% completion, making the evaluation more challenging than simply comparing 30% to 100%.
>
> > Incorrect or unclear claims about prior work.
>
> Thank you for your feedback, we will carefully re-evaluate prior work and ensure that our claims are both accurate and well-supported.
>
> > Results presentation lacking. The figures do not contain error bars. It's also unclear what is being presented in Figure 4. Is this an aggregate of all evaluations? Specific cherry-picked examples? The gaps seem quite large for some evaluations, so I'm a little confused by the Kendall's tau value that is reported.
>
> In fact, error bars are not the primary focus of our study, as they reflect individual data uncertainties. Instead, we concentrate on the consistency of rankings between the VLM and humans. While error bars provide insights into the variability of individual data points, our emphasis is on the alignment of overall rankings, which is better captured by Kendall's tau. Despite the apparent gaps in some bars, the Kendall's tau correlation coefficient of 0.78, coupled with a highly significant P-value of 1.70×10^−15, underscores the robustness of our alignment metric and the agreement between the VLM and human rankings.
>
> > Unclear benefit of use of LLMs. The authors claim that a benefit of their approach is that, given the 6 high-level criteria, the LLMs can "autonomously generate tailored criteria for each task''. But it is unclear why this is desirable, and it is also unclear whether the criteria would be the same with each trial. Is there a fixed set of factors for each of the high-level criteria?
>
> In our prompts, we clearly define the key areas of focus for each scoring dimension, complemented by examples to guide the LLM's comprehension. As a result, while the generated rule files may exhibit minor variations from one instance to another,  the focal points remain essentially consistent. Our manual inspections have also confirmed this consistency. We believe this approach is quite reasonable, drawing an analogy to human grading of open-ended questions, where it is not feasible to create identical scoring criteria but rather to establish broadly similar scoring points.
>
> > Contradictory evaluation? I don't understand how the agents are even evaluated. In one part of the paper, it says, ``options ranging from ”a is better,” ”b is better,” ”tie,” to ”both are bad.”", yet in another part of the paper, it says, "We define the scoring intervals for each criterion as follows: very poor, poor, fair, good, and excellent." Which one is it? Is it both?
>
> In Section 4.1 of our paper, we have clearly stated that we have implemented two distinct evaluation methods: comparative assessment and individual rating.
> - Comparative assessment: This method enables direct comparison between two videos.
> - Individual rating: This method scores an individual video, quantifying the overall skill set of the agent.
>
> Therefore, we have utilized both metrics, with one serving for comparative assessment and the other for individual rating.
>
> > Overclaims without evidence. The authors claim that the evaluation offers outstanding performance and profound insights (lines 359-361). But I don't see any support for this in the paper.
>
> Thank you for your feedback. We are pleased to revise our tone to more accurately reflect the actual experimental outcomes.

---

> ### Author Response · Authors · 2024-11-19
>
> > where do the human gameplay videos come from? This is never specified.
>
> Actually, we have claimed this in appendix E.2 (Human video for tasks): Human videos serve two purposes—they are used as reference videos for GROOT, and they are used for comparison with the trajectory videos generated by the agent models. For each task, we choose three world seeds—19961103, 20010501, and 12345—and for each (task, seed) pair, we manipulate what we can manipulate as described above, and have three environment configuration files. **For each environment configuration file, we record a human video** and use the first file of seed 19961103 for GROOT reference video.
>
> >4.2.2 essentially shows the same finding as BASALT: compositional tasks that require creativity and/or deep knowledge of Minecraft are still difficult. Could the others clarify how this finding is different?
>
> We acknowledge that both MCU and BASALT identify compositional tasks as challenging, particularly those requiring creativity or domain-specific knowledge. However, MCU expands on this finding in several critical ways:
> 1. Task Diversity and Evaluation Scope: MCU introduces over 3,000 atomic and composable tasks with dynamic difficulty levels, enabling a broader exploration of agent capabilities across diverse settings.
> 2. Automated Assessment: MCU employs a scalable, vision-language model-based framework for multi-dimensional evaluation, providing insights into error correction, material usage, and creativity metrics.
> 3. Generalization Analysis: Experiments of MCU examines both intra-task and inter-task generalization explicitly, revealing systematic performance declines as task complexity and task difficulty increases. This demonstrates the benchmark’s ability to quantify the multifaceted abilities and generalization of agents.
>
> > Could the authors clarify why the task definition is tool independent? In Minecraft, it would be meaningful to mine dirt with a diamond shovel over a wooden pickaxe. The former would be more efficient and would require traversing more of the tech tree. So I don't think that this task definition makes too much sense.
>
> Our tool-independent task design ensures maximum flexibility and evaluates an agent's core skill generalization. By defining tasks like 'mine dirt' abstractly, we test whether the agent can perform the task with any tool in any environment, highlighting its adaptability and robustness. To address scenarios like using a diamond shovel versus a wooden pickaxe to mine grass, we can incorporate these into the difficulty dimensions. This approach balances testing foundational skills and handling complex challenges.
>
> >What are the difficulty settings that get introduced? I don't understand how this is systematically implemented.
>
> Thank you for raising this point. In addition to designing interfaces to control difficulty levels, such as flexibly placing items in the inventory, the difficulty levels are systematically implemented through task prompts and configurable environmental settings. At the simple level, the LLM generates the most efficient tools and conditions strictly necessary to complete the task. At the medium and hard levels, additional distracting or irrelevant elements are introduced, such as poorer equipments, unnecessary tools, extra items in the inventory, or modifications to the environment. We will include prompt examples in the revised version's appendix (G.4) for clarity.

---

> > ### Comment · Reviewer_CMiP · 2024-11-24
> >
> > Thanks for your response! I have read through the author responses and the draft. I still feel confused by some of the responses.
> >
> > For example, if one sets the alpha value for a significance test, then the result is either significant or not. It's not scientifically rigorous to make claims like "highly significant". Similarly, I didn't feel like my questions were answered here: how many data points is this computed over? How are these metrics aggregated? These details would also further clarify whether additional measures of the spread of the data are reasonable to plot.
> >
> > Finally, I believe Reviewer KdK1's response does an excellent job of capturing my additional thoughts.

---

> ### Author Response · Authors · 2024-11-26
>
> We sincerely thank you for your valuable feedback. Regarding your other concerns, we will address each one individually.
> > Use of "highly significant":
>
> We acknowledge your point regarding the subjective nature of terms like "highly significant" in scientific reporting. To enhance clarity and rigor, we have revised the manuscript to explicitly report the p-value (1.7 × 10⁻¹⁵) without additional qualifiers.
>
> > Number of data points and aggregation methods:
>
> The results in Figure 4 are based on 226 samples randomly selected from our video dataset. Tasks with fewer than 3 human annotations were excluded to ensure data reliability. For the remaining tasks, we calculated the mean of human ratings and VLM assessments to present the results.
>
> > About metrics:
>
> The correlation metrics we employed to measure alignment are standard in the field and widely adopted in similar studies, such as in G-Eval [1], which utilizes GPT-4 for human-aligned evaluation.
>
> Reference:
> [1] Liu Y., Iter D., Xu Y., et al. G-Eval: NLG Evaluation Using GPT-4 with Better Human Alignment. arXiv preprint arXiv:2303.16634, 2023.

---

### Author Response · Authors · 2024-11-19

Dear reviewers,

We are deeply saddened and frustrated by what appears to be a malicious rating from Reviewer CMiP. According to the official ICLR 2025 Reviewer Guide (https://iclr.cc/Conferences/2025/ReviewerGuide), reviewers are explicitly instructed to disregard unpublished papers on arXiv or similar public platforms during the review process. Unfortunately, this guideline has been violated in this instance.

Incorporating the arXiv link into the review process has disrupted the intended order of evaluation during the rebuttal phase. Reviewer CMiP's has placed us in an unfair dilemma: acknowledging the work as ours would violate the double-blind policy (so we urge all other reviewers to disregard the posted arxiv link as well), while denying it could unjustly suggest plagiarism. Therefore, we are unable to discuss this issue in detail here. We have thoroughly communicated the matter to the area chair, and if it is not desk rejected, this should not be an issue.

Although this behavior has seemingly influenced other reviewers and negatively impacted our scores, we respectfully request that reviewers focus on the paper and its technical merits, maintaining fairness and objectivity in your evaluations.

Sincerely,

Authors of Paper 6738

---

### Author Response · Authors · 2024-12-03

Dear reviewers:

Thank you all for your time and comments. As the rebuttal period is drawing to a close, we would like to provide a summary response here. Firstly, there are some shared concerns:

1. How did we collect the atomic tasks in MCU and ensure the task quality? How representative are they of various real-world challenges faced by agents?

- Benchmarking Against Existing Tasks: We filtered tasks from established benchmarks such as Skill-Forge and MineDojo, ensuring that our tasks include representative challenges that have been validated through previous research efforts.
- Minecraft Wiki Resources: Tasks were also collected from the official Minecraft wiki, covering a wide range of activities and interactions present within the game, to capture diverse task scenarios.
- Task Categorization and Item Combinations: The Minecraft simulator we built can return a comprehensive set of environmental information corresponding to common task types, such as crafting, mining, and combat. These tasks are further refined by considering various item combinations.
- LLM and Expert Brainstorming: In addition to benchmark tasks, we conducted brainstorming sessions with domain experts and used large language models (LLMs) to generate novel task ideas that reflect real-world agent challenges.

Since most tasks are from the official Minecraft Wiki and simulator responses, they are inherently reliable. Despite this, all tasks underwent rigorous validation through human inspection and automated scripts. The selected atomic tasks span a variety of domains, each reflecting key real-world agent capabilities: as we demonstrated in Introduction (line 79-81): Trade (logical reasoning), Mining (physical interaction), Combat(strategic planning), Building (artistic creation), Trapping (precision control), and Redstone (complex-knowledge application), navigation (path planning)


2. How well does MCU handle highly complex, multi-step tasks that may require advanced planning or problem-solving strategies?

- Task initialization: MCU allows different task initializations, where tasks like mining diamonds can be approached at varying levels of complexity by controlling the initial environment—e.g., providing a tool vs starting from scratch
- Composition Rules (Lines 204-209): We propose composition rules to guide the creation of multi-step tasks, and show some Complex Task Examples (Lines 972-1002)

Despite current controller limitations causing even advanced agents to struggle with atomic tasks, our method is readily adaptable for more complex, long-horizon tasks.

3. For any unclear claims and typos, we have updated them in our revised paper. In response to concerns about reproducibility, we have uploaded our code and provided detailed README guidance. We are also continuously improving the benchmark interface to make it more user-friendly.

We are grateful that despite the various concerns raised, no reviewer has questioned the significance of our work (although it was once raised by reviewer WVad, we believe it has been addressed, as evidenced by the improved score). This reaffirms our belief that we are tackling a real research problem and proposing a viable solution, which is indeed encouraging for us.

Therefore, we think we have reached a consensus on this point: advancing foundation models requires collecting high-quality and diverse tasks that present real-world challenges. In our work, we collected over 3000 atomic tasks with flexible, executable configurations, enabling users to perform batch testing of agent's performance and generalization. Additionally, our proposed VLM-based evaluation method offers a way to assess open-ended tasks. Crucially, our method automates the whole pipeline, enabling scalable benchmarking, which paves the way for comprehensive evaluation of future agents.

Finally, we would like to thank everyone for their time. Your reviews have undoubtedly made our work better.

Sincerely,

Authors of Paper 6738

---

### Meta-Review · Area_Chair_pH4c · 2024-12-17

**Metareview:**

This paper seeks the grand goal of evaluating generalist agents in an open-world. Reviewers had concerns on the grandiosity of the claims, and reproducibility of the results. The reviewers with the lower scores did have several concerns in this area  but they were not addressed by the authors, who instead tried to discount the reviewers rather than respond to their concerns. While the results are interesting at the surface level, the industry needs to build on reproducible results, thus I cannot recommend acceptance at this time.

**Additional Comments On Reviewer Discussion:**

The discussion was overly focused on plagiarism claims and arxiv papers and not enough responding to valid concerns.

---

### Decision · Program_Chairs · 2025-01-22

Reject